# For Better or For Worse? Learning Minimum Variance Features With Label Augmentation

**Muthu Chidambaram & Rong Ge**
Department of Computer Science, Duke University

## Abstract

Data augmentation has been pivotal in successfully training deep learning models on classification tasks over the past decade. An important subclass of data augmentation techniques - which includes both label smoothing and Mixup - involves modifying not only the input data but also the input label during model training. In this work, we analyze the role played by the label augmentation aspect of such methods. We first prove that linear models on binary classification data trained with label augmentation learn only the minimum variance features in the data, while standard training (which includes weight decay) can learn higher variance features. We then use our techniques to show that even for nonlinear models and general data distributions, the label smoothing and Mixup losses are lower bounded by a function of the model output variance. Lastly, we demonstrate empirically that this aspect of label smoothing and Mixup can be a positive and a negative. On the one hand, we show that the strong performance of label smoothing and Mixup on image classification benchmarks is correlated with learning low variance hidden representations. On the other hand, we show that Mixup and label smoothing can be more susceptible to low variance spurious correlations in the training data.

## 1 Introduction

The training and fine-tuning procedures for current state-of-the-art (SOTA) computer vision models rely on a number of different data augmentation schemes applied in tandem (Yu et al., 2022; Wortsman et al., 2022; Dehghani et al., 2023). While some of these methods involve only transformations to the input training data - such as random crops and rotations (Cubuk et al., 2019) - a non-trivial subset of them also apply transformations to the input training label.

Perhaps the two most widely applied data augmentation methods in this subcategory are label smoothing (Szegedy et al., 2015) and Mixup (Zhang et al., 2018). Label smoothing replaces the one-hot encoded labels in the training data with smoothed out labels that assign non-zero probability to every possible class (see Section 2 for a formal definition). Mixup similarly smooths out the training labels, but does so via introducing random convex combinations of data points and their labels. As a result, Mixup modifies not only the training labels but also the training inputs. The general principles of label smoothing and Mixup have been extended to several variants, such as Structural Label Smoothing (Li et al., 2020), Adaptive Label Smoothing (Wang et al., 2021), Manifold Mixup (Verma et al., 2019), CutMix (Yun et al., 2019), PuzzleMix (Kim et al., 2020), SaliencyMix (Uddin et al., 2020), AutoMix (Liu et al., 2021), and Noisy Feature Mixup (Lim et al., 2022).

Due to the success of label smoothing and Mixup-based approaches, an important question on the theoretical side has been understanding *when and why* these data augmentations improve model performance. Towards that end, several recent works have studied this problem from the perspectives of regularization (Guo et al., 2019; Carratino et al., 2020; Lukasik et al., 2020; Chidambaram et al., 2021), adversarial robustness (Zhang et al., 2020), calibration (Zhang et al., 2021; Chidambaram & Ge, 2023), feature learning (Chidambaram et al., 2023; Zou et al., 2023), and sample complexity (Oh & Yun, 2023).

Although the connection between the label smoothing and Mixup losses has been noted in both theory and practice (Carratino et al., 2020), there has not been (to the best of our knowledge) a unifying theoretical perspective on why the models trained using these losses exhibit similar behavior. In this paper, we aim to provide such a perspective by extending the ideas from previous feature learning analyses to the aspect shared in common between both Mixup and label smoothing: label

augmentation. Our analysis shows that both Mixup and label smoothing hone in on low variance features in the data, and we demonstrate empirically that this phenomenon occurs in both synthetic settings with spuriously correlated features as well as in the training of standard deep learning models for image classification benchmarks.

## 1.1 MAIN CONTRIBUTIONS

The main message of our paper can be summarized as:

> *In data distributions where there are low variance and high variance latent features, both label smoothing and Mixup incentivize learning models that correlate more strongly with the low variance features.*

We prove this message concretely in Section 3.1, where we characterize linear optimizers of the label smoothing and Mixup losses on data in which some dimensions have lower variance than others. In Section 3.2, we also prove weaker analogues of our linear results in the context of general models and multi-class distributions; namely, we show that optimizing the label smoothing and Mixup losses requires decreasing model output variance in a way that is not necessarily true for empirical risk minimization (ERM) combined with weight decay.

We verify our theory empirically in Section 4, where we show that label smoothing and Mixup do indeed learn lower variance hidden representations (corroborating ideas from the literature on neural collapse (Papyan et al., 2020)) and that this lower variance property *correlates with better generalization performance*. We hypothesize that, for standard benchmarks, there exist **low variance latent features that generalize well** as opposed to high-variance, noisy features, and we expect that directly investigating such features would be a fruitful line of future work. However, we also point out that our key message does not directly imply better generalization performance – we show via synthetic experiments in Section 4.2 that it is possible for the low variance features to actually be spuriously correlated with the targets (e.g. fixed backgrounds).

## 1.2 RELATED WORK

**Label Smoothing.** Since being introduced by Szegedy et al. (2015), label smoothing continues to be used to train SOTA vision (Wortsman et al., 2022; Liu et al., 2022), translation (Vaswani et al., 2017; Team et al., 2022), and multi-modal models (Yu et al., 2022). Attempts to understand when and why label smoothing is effective can be traced back to Pereyra et al. (2017) and Müller et al. (2020), which respectively relate label smoothing to entropy regularization and show that it can lead to more closely clustered learned representations. Müller et al. (2020) also show that label smoothing can improve calibration but hurt distillation performance; Zhu et al. (2023); Xia et al. (2024) further show that these improvements in calibration do not translate to better selective classification.

On the theoretical side, Lukasik et al. (2020) study the relationship between label smoothing and loss correction techniques used to handle label noise, and show that label smoothing can be effective for mitigating label noise. Liu (2021) extends this line of work by analyzing how label smoothing can outperform loss correction in the context of memorization of noisy labels. Wei et al. (2022) further show that negative label smoothing can outperform traditional label smoothing in the presence of label noise. Xu et al. (2020) provide an alternative theoretical perspective, showing that label smoothing can improve convergence of stochastic gradient descent.

**Mixup.** Similar to label smoothing, Mixup (Zhang et al., 2018) and its aforementioned variants have also played an important role in the training of SOTA vision (Wortsman et al., 2022; Liu et al., 2022), text classification (Sun et al., 2020), translation (Li et al., 2021), and multi-modal models (Hao et al., 2023), often being applied alongside label smoothing. Initial work on understanding Mixup studied it from the perspective of introducing a data-dependent regularization term to the empirical risk (Carratino et al., 2020; Zhang et al., 2020; Park et al., 2022), with Zhang et al. (2020) and Park et al. (2022) showing that this regularization effect can lead to improved Rademacher complexity.

On the other hand, Guo et al. (2019) and Chidambaram et al. (2021) show that the regularization terms introduced by the Mixup loss can also have a negative impact due to the fact that Mixup-augmented points may coincide with existing training data points, leading to models that fail to minimize the original risk. Mixup has also been studied from the perspective of calibration (Thulasidasan et al., 2019), with theoretical results (Zhang et al., 2021; Chidambaram & Ge, 2023) showing that Mixup training can prevent miscalibration issues arising from ERM.

Recently, Oh & Yun (2023) studied Mixup in a similar context to our work (binary classification with linear models), and showed that Mixup can significantly improve the sample complexity required to learn the optimal classifier when compared to ERM. Most closely related with our results in this paper, Mixup has been studied from a feature learning perspective, with Chidambaram et al. (2023) and Zou et al. (2023) both showing that Mixup training can learn multiple features in a generative, multi-view (Allen-Zhu & Li, 2021) data model despite ERM failing to do so.

**Neural Collapse.** Papyan et al. (2020) showed that the last layer activations of trained neural networks collapse to their class means, in such a way that they are maximally separable. Kornblith et al. (2021); Zhou et al. (2022); Guo et al. (2024) all investigate the interplay between label smoothing and this effect, with Kornblith et al. (2021) showing that label smoothing can lead to more separable last layer representations (although worse linear transfer performance) and Zhou et al. (2022); Guo et al. (2024) showing that it can also lead to faster convergence to these collapsed representations. We corroborate these results with our experiments in Section 4.1, where we show that both Mixup and label smoothing lead to much lower total variance in last layer activations compared to standard cross-entropy.

**General Data Augmentation.** General data augmentation techniques have been a mainstay in image classification since the rise of deep learning models for such tasks (Krizhevsky et al., 2012). As a result, there is an ever-growing body of theory (Bishop, 1995; Dao et al., 2019; Wu et al., 2020; Hanin & Sun, 2021; Rajput et al., 2019; Yang et al., 2022; Wang et al., 2022; Chen et al., 2020; Mei et al., 2021) aimed at addressing broad classes of augmentations such as those resulting from group actions (i.e. rotations and reflections). Recently, Shen et al. (2022) also studied a class of linear data augmentations from a feature learning perspective, once again using a data model based on the multi-view data of Allen-Zhu & Li (2021). This work is in the same vein as that of Zou et al. (2023) and Chidambaram & Ge (2023), although the augmentation considered is not comparable to Mixup.

## 2 PRELIMINARIES

**Notation.** Given $n \in \mathbb{N}$, we use $[n]$ to denote the set $\{1, 2, ..., n\}$. For a vector $x \in \mathbb{R}^d$ and a subset $S \subset [d]$, we use $x_S \in \mathbb{R}^{|S|}$ to denote the restriction of $x$ to only those indices in $S$, and also use $x_i$ to denote the $i^{\text{th}}$ coordinate of $x$. The same notation also applies to matrices; i.e. for a square matrix $T$ we use $T_S$ to denote the restriction of both the rows and columns of $T$ to only those dimensions in $S$. We use $\mathrm{I}_d$ to denote the identity matrix in $\mathbb{R}^d$. Additionally, we use $\succ$ to denote the partial order over positive definite matrices and $\mathrm{I}_d$ to denote the identity matrix. We use $\|\cdot\|$ to indicate the Euclidean norm on $\mathbb{R}^d$. For a function $g : \mathbb{R}^n \to \mathbb{R}^m$ we use $g^i(x)$ to denote the $i^{\text{th}}$ coordinate function of $g$. We use $\Delta^{k-1}$ to denote the $(k-1)$-dimensional probability simplex in $\mathbb{R}^k$. For a probability distribution $\pi$ we use $\mathrm{supp}(\pi)$ to denote its support. Additionally, if $\pi$ corresponds to the joint distribution of two random variables $X$ and $Y$ (i.e. data and label), we use $\pi_X$ and $\pi_Y$ to denote the respective marginals, and $\pi_{X|Y=y}$ and $\pi_{Y|X=x}$ to denote the regular conditional distributions. We use $\Sigma_X$ to denote the covariance matrix of a random variable $X \in \mathbb{R}^d$. Lastly, we use $\mathrm{Var}(X)$ to denote $\mathrm{Tr}(\Sigma_X)$ for $X \in \mathbb{R}^d$.

We consider the $k$-class classification setting, in which there is a ground-truth data distribution $\pi$ on $\mathbb{R}^d \times [k]$ and our goal is to model the conditional distribution $\pi_{Y|X}$ using a learned function $g$. In our main theoretical results, we will pay particular attention to the case where $k = 2$ and $g$ is a logistic regression model (although we generalize a weak version of these observations to $k$ classes as well), as in this setting we can get a clear handle on the features in the data learned by an optimal model with respect to a particular loss. For this case, we will assume that $\pi$ is supported on $\mathbb{R}^d \times \{\pm 1\}$ and that $g$ is parameterized by a weight vector $w$, i.e. $g_w(x) = \sigma(w^\top x)$, where $\sigma$ is the sigmoid function. Throughout this work, we will consider the following three families of losses.

**Standard cross-entropy with optional weight decay.** The canonical cross-entropy (or negative log-likelihood) objective in the $k$-class setting is defined as:

$$\ell(g) = \mathbb{E}_{(X,Y)\sim\pi}[-\log g^Y(X)]. \tag{2.1}$$

Here we have not specified a weight decay term, since we have placed no constraints on the structure of $g$. On the other hand, for linear binary classification, we can define the binary cross-entropy with optional weight decay as (recalling that $Y \in \{\pm 1\}$)

$$\ell_\beta(w) = \mathbb{E}_{(X,Y)\sim\pi}\left[-\log g_w(YX)\right] + \frac{\beta}{2}\|w\|^2, \tag{2.2}$$

where we have defined the loss in terms of the parameter vector $w$. When $\beta > 0$, (2.2) has a unique minimizer and we will directly analyze its properties. On the other hand, when $\beta = 0$ (no weight decay), this is no longer the case - but our results will still apply to the common case of minimizing (2.2) by scaling the max-margin solution, which we define below.

**Definition 2.1.** [Max-Margin Solution] The max-margin solution $w^*$ with respect to $\pi$ is defined as:

$$w^* = \underset{w \in \mathbb{R}^d}{\text{argmin}} \, \|w\|^2$$

$$\text{s.t. } y \langle w, x \rangle \geq 1 \quad \text{for } \pi\text{-a.e.}(x, y). \tag{2.3}$$

**Label smoothing.** The cross-entropy as defined in (2.1) treats the reference distribution that we compare $g(X)$ to as a point mass on the class $Y$. On the other hand, the label-smoothed cross-entropy is obtained by instead treating the reference distribution as a mixture of a point mass on $Y$ and the uniform distribution over $[k]$. Namely, the label-smoothed loss with mixing hyperparameter $\alpha \in [0, 1]$ is defined to be:

$$\ell_{\text{LS},\alpha}(g) = -\mathbb{E}_{(X,Y)\sim\pi}\left[(1-\alpha)\log g^Y(X) + \frac{\alpha}{k}\sum_{i=1}^{k}\log g^i(X)\right]. \tag{2.4}$$

And the corresponding binary version is (once again, $Y \in \{\pm 1\}$):

$$\ell_{\text{LS},\alpha}(w) = -\mathbb{E}_{(X,Y)\sim\pi}\left[\left(1 - \frac{\alpha}{2}\right)\log g_w(YX) + \frac{\alpha}{2}\log g_w(-YX)\right]. \tag{2.5}$$

**Mixup.** Similar to label smoothing, Mixup also augments the reference distribution from being a point mass on $Y$ to being a mixture. However, Mixup also augments the input data as well. In particular, Mixup considers convex combinations of two pairs of points $(X_1, Y_1)$ and $(X_2, Y_2)$, with a mixing weight sampled from a distribution $\mathcal{D}_\lambda$ (which is a hyperparameter) whose support is contained in $[0, 1]$. To simplify notation, we will use $X_{1:n}$ and $Y_{1:n}$ to denote multiple inputs $X_1, ..., X_n$ and their corresponding labels $Y_1, ..., Y_n$, and additionally introduce a function $h$ defined as:

$$h(\lambda, g, X_{1:2}, Y_{1:2}) = \lambda \log g^{Y_1}(Z_\lambda) + (1-\lambda)\log g^{Y_2}(Z_\lambda) \tag{2.6}$$

$$\text{where} \quad Z_\lambda = \lambda X_1 + (1-\lambda)X_2. \tag{2.7}$$

After which we can define the Mixup cross-entropy as:

$$\ell_{\text{MIX},\mathcal{D}_\lambda}(g) = \mathbb{E}_{(X_{1:2},Y_{1:2})\sim\pi\otimes\pi,\lambda\sim\mathcal{D}_\lambda}\left[-h(\lambda, g, X_{1:2}, Y_{1:2})\right]. \tag{2.8}$$

The corresponding binary version $\ell_{\text{MIX},\mathcal{D}_\lambda}(w)$ of (2.8) is identical except for redefining $h$ to be:

$$h(\lambda, w, x_{1:2}, y_{1:2}) = \lambda \log g_w(Y_1 Z_\lambda) + (1-\lambda)\log g_w(Y_2 Z_\lambda). \tag{2.9}$$

## 3 MAIN THEORETICAL RESULTS

All omitted proofs in this section can be found in Section A of the Appendix.

### 3.1 LINEAR BINARY CLASSIFICATION

We begin first with the binary setting, in which we will consider a data model where a subset of the input dimensions correspond to a *low variance feature* and the complementary dimensions correspond to a *high variance feature* that is more separable. We emphasize that our notion of "feature" here does not correspond to explicit feature vectors that are fixed per class like in prior work, we simply **designate subsets of the dimensions as features** for simplicity. This data model can be interpreted in multiple ways: depending on the context, we may wish to learn either both of the features present in the data or simply hone in on the low variance feature (for example, identifying a stop sign with many different backgrounds).

Although this setting seems simplistic – one may object that the variance differences in the input can be handled with suitable normalization/whitening – the insights from our data model can be applied to *learned features* (i.e. intermediate representations in a deep learning model) where such modifications are less straightforward, as we demonstrate in the experiments of Section 4.

Our main results in this section show that doing any kind of label augmentation (label smoothing or Mixup) in our low variance/high variance feature setup will lead to a model that has only learned the low variance feature, whereas minimizing the binary cross-entropy with non-zero weight decay requires learning the high variance feature due to its greater separability.

**Definition 3.1.** [Binary Data Distribution] We consider $\pi$ to be a distribution supported on $K \times \{\pm 1\}$ where $K$ is a compact subset of $\mathbb{R}^d$. We assume that $\pi$ is nondegenerate in that it satisfies $\pi_Y(y) > 0$ for each $y \in \{\pm 1\}$ and that $\Sigma_X$ is positive definite, and we also assume $\mathbb{E}[X] = 0$. Additionally, we designate a subset $\mathcal{L} \subseteq [d]$ that we refer to as the low variance feature in the data, and we refer to the complement $\mathcal{H} = \mathcal{L}^c$ as the high variance feature.

For convenience, for a vector $v \in \mathbb{R}^d$, we will use $v \in \mathcal{L}$ to mean that only the $\mathcal{L}$ dimensions of $v$ are non-zero. Definition 3.1 only treats $\mathcal{L}$ and $\mathcal{H}$ as placeholders; this is because our weight decay result is insensitive to variance assumptions and only depends on differences in the separability of the dimensions $\mathcal{L}$ and $\mathcal{H}$, as we indicate below.

**Assumption 3.2.** We assume that for every unit vector $u^* \in \mathcal{L}$, there exists a unit vector $v^* \in \mathcal{H}$ and $y \langle v^*, x \rangle > y \langle u^*, x \rangle$ for $\pi$-a.e. $(x, y)$.

Assumption 3.2 is strong, but it actually does not imply linear separability, only that the $\mathcal{H}$ dimensions are in a sense better than the $\mathcal{L}$ dimensions for classification. We explore a simple 2-D distribution illustrating Assumption 3.2 in Appendix C.1.

Our first result shows that for $\ell_\beta(w)$ as defined in (2.2), the minimizer $w^*$ has a large correlation with the dimensions in $\mathcal{H}$. This is of course intuitive given Assumption 3.2, but it is not immediate because the weight decay penalty in (2.2) encourages distributing norm across all of the dimensions of $w$ (e.g. $\sum w_i$ is maximized with respect to the constraint $\|w\| = 1$ by considering $w_i = 1/\sqrt{d}$).

**Theorem 3.3.** *Let $w^*$ be the unique minimizer of $\ell_\beta(w)$ for $\beta > 0$ under $\pi$ satisfying Assumption 3.2. Then $\|w^*_\mathcal{H}\|^2 \geq \frac{1}{2}\|w^*\|^2$.*

**Proof Sketch.** We can orthogonally decompose the optimal solution $w^*$ in terms of unit normal directions $u^* \in \mathcal{L}$ and $v^* \in \mathcal{H}$. We show that in this decomposition it must be the case that $y \langle v^*, x \rangle > y \langle u^*, x \rangle$ for $\pi$-a.e. $(x, y)$, and then we can claim that $w^*$ must have greater weight associated with $v^*$ than $u^*$, as otherwise we can decrease $\ell_\beta(w^*)$ by moving weight from $u^*$ to $v^*$.

We cannot immediately extend the result of Theorem 3.3 to the binary cross-entropy $\ell_0(w)$ without weight decay, since there is no unique minimizer of $\ell_0$. However, for linear models trained with gradient descent (as is often done in practice), it is well-known that the learned model converges in direction to the max-margin solution (Soudry et al., 2018; Ji & Telgarsky, 2020). For this case, the proof technique of Theorem 3.3 readily extends and we obtain the following corollary.

**Corollary 3.4.** *If $w^*$ is the max-margin solution to $\ell_0(w)$, then the result of Theorem 3.3 still holds.*

On the other hand, we will now show that once we introduce variance assumptions on $\mathcal{L}$ and $\mathcal{H}$, this phenomenon does not occur when minimizing the label smoothing and Mixup losses $\ell_{\mathrm{LS},\alpha}$ and $\ell_{\mathrm{MIX},\mathcal{D}_\lambda}$. In both cases, the optimal solutions have arbitrarily small correlation with the dimensions in $\mathcal{H}$ as the distribution of $YX$ concentrates.[1] For these results, we require the following assumptions (but no longer need Assumption 3.2).

**Assumption 3.5.** We assume that $\mathbb{E}[YX_\mathcal{L}] \neq 0$ and $\Sigma_{YX,\mathcal{H}} \succ \rho \mathrm{I}_d$ for some $\rho > 0$. Here $\Sigma_{YX,\mathcal{H}}$ denotes the covariance matrix of $\Sigma_{YX}$ restricted to those rows and columns in $\mathcal{H}$.

The first part of Assumption 3.5 just ensures it is possible to obtain a good solution using the dimensions in $\mathcal{L}$ while the second part codifies the idea of $\mathcal{H}$ being a high (at least non-zero) variance feature. Observe that we have made *no direct separability assumptions on the data*, although it is true that as $\|\Sigma_{YX,\mathcal{L}}\| \to 0$ the class-conditional supports are guaranteed to be linearly separable in $\mathcal{L}$. This means that we can consider settings where both Assumptions 3.2 and 3.5 are true, and in such settings there will be a clear separation between weight decay and label smoothing/Mixup. This kind of separation in the learned decision boundaries is visualized in Appendix C.1, which provides intuition for the next two results.

**Theorem 3.6.** *For $\alpha \in (0, 1)$ and $\pi$ satisfying Assumption 3.5, every minimizer $w^*$ of $\ell_{\mathrm{LS},\alpha}(w)$ satisfies $\|w^*_\mathcal{H}\| < O(\|\Sigma_{YX,\mathcal{L}}\|)$.*

---

[1]In order for the variance of $YX$ to go to 0, we need the classes to be balanced; however, we can change our assumptions to be in terms of the conditional variance of $X$ to remove this requirement.

**Proof Sketch.** We show that $\ell_{\text{LS},\alpha}(w)$ is strongly convex in $w^\top YX$. We then use Jensen's inequality to get a lower bound on the loss in terms of this quantity, and show that this lower bound is achievable using only $w \in \mathcal{L}$ as the variance of $YX_{\mathcal{L}}$ decreases. Then using a lower bound on the Jensen gap (Lemma A.1) and Assumption 3.5, we show that any solution that is sufficiently non-zero in the $\mathcal{H}$ dimensions remains bounded away from this lower bound.

**Theorem 3.7.** *For any symmetric $\mathcal{D}_\lambda$ that is not a point mass on $0$ or $1$ and $\pi$ satisfying Assumption 3.5, every minimizer $w^*$ of $\ell_{\text{MIX},\mathcal{D}_\lambda}(w)$ satisfies $\|w_{\mathcal{H}}^*\| < O(\|\Sigma_{YX,\mathcal{L}}\|)$.*

**Proof Sketch.** Mixup differs from label smoothing in that we need to condition on $\lambda$ and then show strong convexity of the conditional loss in terms of $w^\top Z_\lambda$. We can then get a lower bound similar to the label smoothing case, but it is no longer immediate that there exists a stationary point $w \in \mathcal{L}$ that minimizes this lower bound. We prove the existence of such a stationary point by considering limiting behavior of the gradient of the loss as $w_{\mathcal{L}}$ tends to $\infty$ or $-\infty$ in each component, after which the rest of the proof follows the label smoothing proof.

## 3.2 GENERAL MULTI-CLASS CLASSIFICATION

Our results so far have demonstrated a separation between standard training (ERM + weight decay) and label augmentation (label smoothing and Mixup) in the linear binary classification setting, without explicitly having to assume linear separability. The benefit of the linear binary case is that in this case the learning problems are convex in the weight vector $w$, so we can directly discuss properties of the optimal solutions instead of worrying about optimization dynamics.

Of course, this is no longer true when we pass to nonlinear models, and standard model choices (e.g. neural networks) make it so that it is no longer easy to prove that a model has "learned" either $X_{\mathcal{L}}$ or $X_{\mathcal{H}}$ without explicitly analyzing some choice of optimization algorithm (i.e. gradient descent). However, our observations do translate to the *outputs* of any model.

It is obvious that the model output variance should go to zero for any model that achieves the global optimum of the label augmentation losses we have discussed; indeed, this just corresponds to predicting the correct labels ($\alpha, 1 - \alpha$ with label smoothing and $\lambda, 1 - \lambda$ for every $\lambda$ for Mixup) with probability 1. On the other hand, it is less clear that we can make a quantitative statement regarding how bad the loss could be given a certain amount of variance in the model output.

By lifting the techniques of the previous subsection, we can prove such quantitative results for both label smoothing and Mixup in the general multi-class setting. The general data distribution we consider for these results is as follows.

**Definition 3.8.** [Multi-Class Data Distribution] We consider $\pi$ to be a distribution supported on $B \times [k]$ where $B$ is a compact subset of $\mathbb{R}^d$ and $k > 2$. We assume only that $\pi$ satisfies $\pi_Y(y) > 0$ for every $y$ and $\Sigma_X$ is positive definite.

We make virtually no assumptions on $\pi$ for these results because we will only be proving general lower bounds, which are weaker than the claims regarding the optimal solutions of the previous subsection. For $\pi$ as in Definition 3.8, the label smoothing and Mixup results are as follows.

**Proposition 3.9.** *For $\alpha > 0$ and any $g : \mathbb{R}^d \to \Delta^{k-1}$, letting $\text{OPT}_{\text{LS},\alpha}$ denote the minimum of $\ell_{\text{LS},\alpha}$, we have for a universal constant $C > 0$:*

$$\ell_{\text{LS},\alpha}(g) \geq \text{OPT}_{\text{LS},\alpha} + C \sum_{y=1}^{k} \pi_Y(y) \text{Var}\left(\mathbb{E}[g(X) \mid Y = y]\right). \tag{3.1}$$

**Proposition 3.10.** *For any $\mathcal{D}_\lambda$ that is not a point mass on $0$ or $1$ and any $g : \mathbb{R}^d \to \Delta^{k-1}$, letting $\text{OPT}_{\text{MIX},\mathcal{D}_\lambda}$ denote the minimum of $\ell_{\text{MIX},\mathcal{D}_\lambda}$, we have for a universal constant $C > 0$ that:*

$$\ell_{\text{MIX},\mathcal{D}_\lambda}(g) \geq \text{OPT}_{\text{MIX},\mathcal{D}_\lambda} + C \sum_{y_1=1}^{k} \sum_{y_2=1}^{k} \pi_Y(y_1)\pi_Y(y_2)\mathbb{E}_\lambda\left[\text{Var}\left(\mathbb{E}[g(Z_\lambda) \mid y_1, y_2, \lambda]\right)\right]. \tag{3.2}$$

The proofs for both Propositions 3.9 and 3.10 follow the same structure; we show that after appropriate conditioning, both losses can be broken up into a sum of strongly convex conditional losses. The lower bounds in both results show that in order to make progress with respect to the label smoothing or Mixup losses, a training algorithm needs to push model outputs to be low variance.

*Remark* 3.11. We do not prove an analogous result to Theorem 3.3 in the general setting because Propositions 3.9 and 3.10 operate directly on the model outputs $g(X)$, and it is not clear how to translate a weight norm constraint to this setting without explicitly parameterizing $g$. Intuitively, however, with a weight norm constraint on $g$ it may no longer be optimal to have zero variance predictions, since such predictions may require very large weights depending on the data distribution.

**Comparisons to existing results.** We are not aware of any existing results on feature learning for label smoothing; prior theoretical work largely focuses on the relationship between label smoothing and learning under label noise, which is orthogonal to the perspective we take in this paper. Although feature learning results exist for Mixup (Chidambaram et al., 2023; Zou et al., 2023), the pre-existing results are constrained to the case of mixing using point mass distributions and only prove a separation between Mixup and unregularized ERM, whereas our results work for arbitrary symmetric mixing distributions (that don't coincide with ERM) and also separate Mixup from ERM with weight decay. This greater generality comes at the cost of considering only linear models for our feature learning results; both Chidambaram et al. (2023) and Zou et al. (2023) consider the training dynamics of 2-layer neural networks on non-separable data.

Additionally, our results can be viewed as generalizing the observations made by Chidambaram et al. (2023), which were that Mixup can learn multiple features in the data when doing so decreases the variance of the learned predictor. In our case, we directly show that Mixup will have much larger correlation with the low variance feature in the data as opposed to the high variance feature. Our results also do not contradict the observations of Zou et al. (2023), which were that Mixup can learn both a "common" feature and a "rare" feature in the data; in their setup the common/rare features are fixed (zero variance) per class and concatenated with high variance noise. We provide a more detailed review of the settings of Chidambaram et al. (2023) and Zou et al. (2023) in Appendix B.

## 4 EXPERIMENTS

We now address the practical ramifications of our theoretical results. In Section 4.1, we show that the intermediate representations learned by deep learning models trained with label smoothing and Mixup do indeed exhibit significantly lower variance when compared to those learned using just weight decay, and that these lower variance representations correlate with better performance. We also show, however, that this minimum variance feature learning can be a detriment by analyzing spurious correlations in the training data in Section 4.2. We also include synthetic experiments directly verifying our theoretical setting in Appendix C.2.

All experiments in this section were conducted on a single A5000 GPU using PyTorch (Paszke et al., 2019) for model implementation. All reported results correspond to means over 5 training runs, and shaded regions in the figures correspond to 1 standard deviation bounds.

### 4.1 LEARNED LOW VARIANCE FEATURES IN IMAGE CLASSIFICATION

We first consider image classification on the standard benchmarks of CIFAR-10 and CIFAR-100 (Krizhevsky, 2009) using ResNets (He et al., 2015); we show results for ResNet-18 in this section and relegate further experiments for deeper architectures to Appendix C.3 (they follow the same trends). We compare the performance of training using just ERM + weight decay to that of ERM + weight decay combined with label smoothing or Mixup; final test error performance is shown in Figure 1 (a) and (d).

Due to compute constraints, we focus on known well-performing settings (Zhang et al., 2018; Müller et al., 2020) for weight decay, label smoothing, and Mixup. Namely, we take the weight decay parameter to be $5 \times 10^{-4}$, the label smoothing $\alpha$ parameter to be $0.1$, and the mixing distribution for Mixup to be $\text{Beta}(1, 1)$ (uniform distribution). We train all models for 200 epochs using a batch size of 1024, a fixed learning rate of $10^{-3}$, and AdamW for optimization (Loshchilov & Hutter, 2019). We preprocess the training data to have zero mean and unit variance along each channel, and also include random crop and flip augmentations as is standard practice for achieving good performance.

Figure 1 shows that adding in label smoothing and Mixup to the baseline of ERM + weight decay leads to noticeable performance improvements, which corroborates existing results in the literature. The novel aspect of our results is shown in Figure 1 (b), (c), (d), and (e), where we track the mean total variance of the penultimate layer activations and output probabilities of each model on the test data over the course of training. The variance values are computed by first computing the total

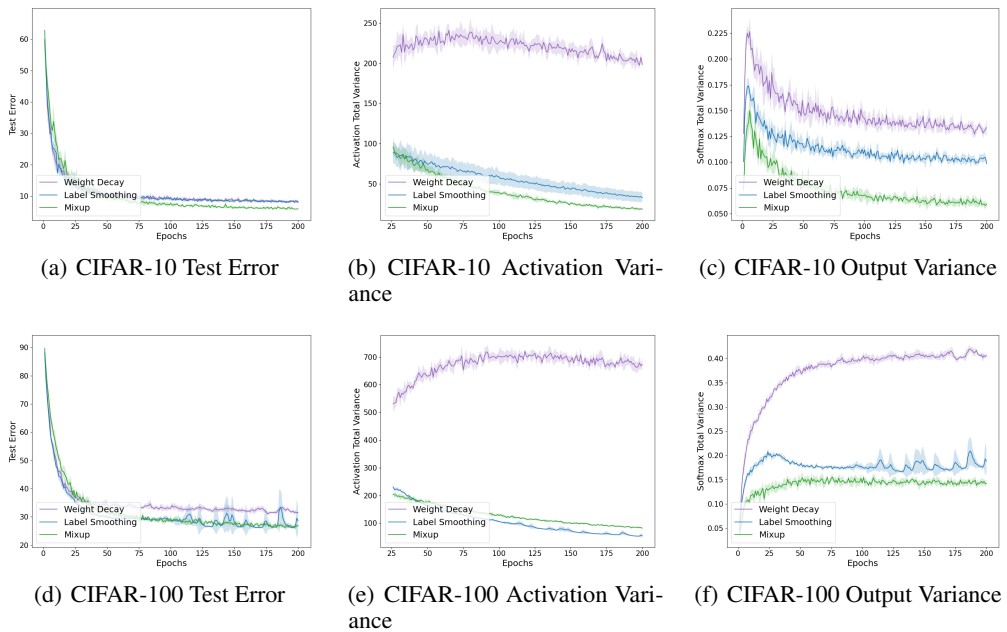

(a) CIFAR-10 Test Error  (b) CIFAR-10 Activation Variance  (c) CIFAR-10 Output Variance

(d) CIFAR-100 Test Error  (e) CIFAR-100 Activation Variance  (f) CIFAR-100 Output Variance

Figure 1: ResNet-18 final test errors, penultimate layer activation variances, and output probability variances on CIFAR-10 and CIFAR-100. Activation variance results are shown starting at epoch 25 as early epochs have larger scale oscillations in the computed variance.

variance of the activations/probabilities (i.e. the sum of the variance in each dimension) *for each class*, and then averaging over all classes. Final test error and variance values are provided in Table 1 and 2 in Appendix C.3.1.

Essentially, we measure the average spread of activations and probabilities across classes, which is similar in spirit to what was done by Müller et al. (2020), although we directly look at variances across all classes whereas they look at projected clusters of a few specific classes. The most telling results are the model activation variances, since it is to an extent expected that per-class variance of model outputs should decrease with improvements in test error (this also corresponds to improved model calibration, which is a known consequence of training with label smoothing and Mixup). That being said, we further analyze model output variance in Appendix C.4 and show that label smoothing and Mixup decrease total output variance by a larger extent than what can be explained by changes in the target class prediction variance alone.

Overall, our results show that adding either label smoothing or Mixup to the baseline of just weight decay leads to *significant* decreases in the activation variances, adding credence to the idea that both methods learn low variance features. We note, however, that our results only establish a correlation between this low variance property and better test performance – it would require a significantly more in-depth empirical study to assess a causal relationship between this kind of feature learning and generalization, which we think would be a fruitful avenue for future work.

### 4.2 Low Variance Spurious Correlations

We now demonstrate that honing in on low variance features can also be *harmful* to performance via binary classification and multi-class classification tasks in which the training data is modified to have spurious correlations with the target. The introduced spurious correlations are *much lower magnitude* than the rest of the data, and in that sense they intuitively satisfy both Assumptions 3.2 and 3.5.

#### 4.2.1 Binary Classification With Perturbed Training Data

For our binary classification task, we consider reductions of the CIFAR benchmarks to binary classification by fixing two classes from each dataset as the positive and negative classes, and replacing their original labels with the labels $+1$ and $-1$ respectively. Our experiments are not

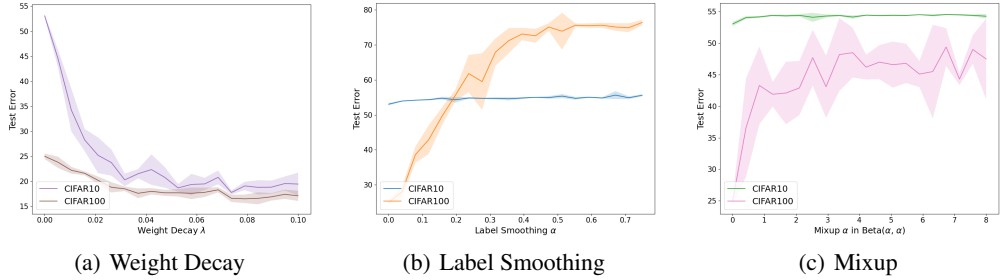

(a) Weight Decay  (b) Label Smoothing  (c) Mixup

Figure 2: Logistic regression final test errors for various hyperparameter settings on the binary classification versions of CIFAR-10 and CIFAR-100 from Section 4.2.1.

sensitive to the choice of binary reduction, and for the experiments in this section we fix the $-1$ class to be class 0 from the original data and the $+1$ class to be class 1 from the original data.

We preprocess the training data to have mean zero and variance one along every image channel, but do not use other augmentations or perform feature extraction using pretrained models. We then "adversarially" modify the training data such that the first value in the tensor representation of each training input is replaced by $\gamma y$ with $\gamma = 0.1$ ($\gamma$ here just needs to be small, we verified the results for $\gamma = 10^{-5}$ up to $\gamma = 0.1$). This ensures that the training data is linearly separable in the first dimension of the data, but learning this first dimension requires having larger weight norm due to the scaling by $\gamma$. We leave the **test data unchanged** - our goal is to determine whether models trained on the modified training data can learn more than just the single identifying dimension.

We then train logistic regression models on both the reduced CIFAR-10 and CIFAR-100 tasks across a range of settings for weight decay, label smoothing, and Mixup. We consider 20 uniformly spaced values in $[0, 0.1]$ for the weight decay $\lambda$ parameter and in $[0, 0.75]$ for the label smoothing $\alpha$ parameter, where the upper bound for the label smoothing parameter is obtained from the experiments of Müller et al. (2020). For Mixup, we fix the mixing distribution to be the canonical choice of $\mathrm{Beta}(\alpha, \alpha)$ introduced by Zhang et al. (2018) and consider 20 uniformly spaced $\alpha$ values in $[0, 8]$ (with $\alpha = 0$ corresponding to ERM), which effectively covers the range of Mixup hyperparameter values used in practice. Other hyperparameters are the same as before, except we use a learning rate of $5 \times 10^{-3}$.

The results across hyperparameter values are shown in Figure 2. Both the label smoothing and Mixup models have high test error for all settings while weight decay achieves a significantly lower test error for all $\lambda > 0$. ERM also has high test error; in this case we differ from the setting of implicit bias results due to training with Adam and likely not training for the period required for convergence to the max-margin solution. Furthermore, these results are insensitive to introducing a small amount of weight decay to the label smoothing and Mixup models (i.e. $5 \times 10^{-4}$ as in the previous subsection).

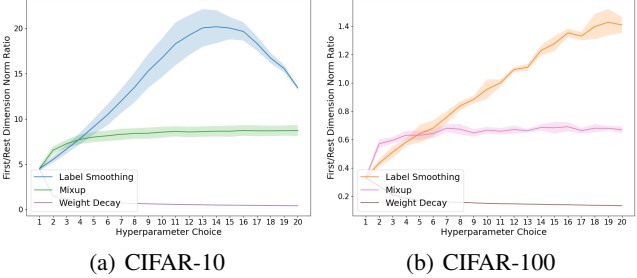

(a) CIFAR-10  (b) CIFAR-100

Figure 3: Comparison of norm ratio between first dimension (synthetically modified in the training data) and remaining dimensions (left unchanged) for trained logistic regression weight vector for the 20 different hyperparameter settings of weight decay, label smoothing, and Mixup.

Our results correspond to label smoothing and Mixup learning to use only the spurious, identifying dimension of the training inputs, as we expect from our theory since this dimension has zero

conditional variance. Indeed, we verify this fact empirically in Figure 3, where we plot the ratio $\|w_1\|/\|w_{[d]\setminus\{1\}}\|$ (i.e. the ratio between the norm of the trained model weight vector in the first dimension and the remaining dimensions) for each of the trained models.

### 4.3 SPURIOUS BACKGROUND CORRELATIONS

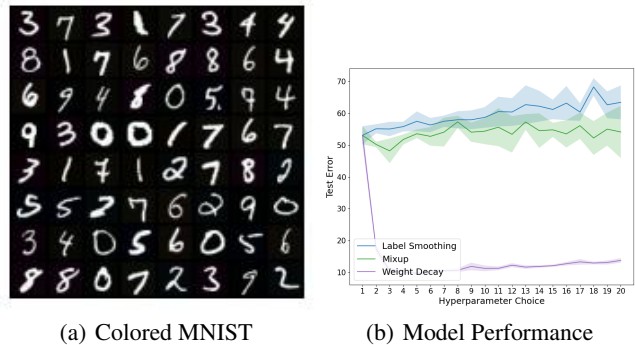

(a) Colored MNIST       (b) Model Performance

Figure 4: Final model test errors over our hyperparameter sweep for the colored MNIST dataset of Section 4.3, alongside a visualization of samples from the dataset.

For our multi-class analogue to Section 4.2.1, we consider a similar setup to the one used by Arjovsky et al. (2020) to motivate the influential invariant risk minimization framework: namely, we construct a colored version of the MNIST dataset in which the background pixels for each class are replaced with different colors corresponding to the class labels. Unlike Arjovsky et al. (2020), however, we maintain all 10 classes since our theory from Section 3.2 suggests that our observations should generalize to the multi-class setting. The colors used for each class background are permuted between the train and the test data, so that a model that learns to predict only using background pixels cannot achieve good test accuracy.

The only constraints we place on the background colors are that they are distinct across classes and that their intensities (i.e. values in RGB space) are small (it suffices to consider values bounded by 16). The latter constraint is not necessary for the failure of label smoothing and Mixup, but is necessary for weight decay to succeed since it ensures that the higher variance feature (the actual digit in the foreground) is generally more separable (due to larger pixel intensities). Note that although there is no variance in the background color conditional on a class, there is significant variance in the actual pixels per class as the locations of the background pixels change across data points.

For our model setup, we also follow Arjovsky et al. (2020) and consider a simple 2-layer feedforward neural network with ReLU activations and a hidden layer size of 2048, as this is sufficient to achieve good performance on MNIST and is a tiny enough model that we can efficiently do the same hyperparameter sweep from Section 4.2.1. Training details remain the same as in Section 4.2.1, except we only train for 20 epochs as this is sufficient for the training loss to roughly converge and greatly expedites the hyperparameter sweep.

Test error results across the different hyperparameter settings for weight decay, label smoothing, and Mixup are shown in Figure 4. We observe the same phenomena as before, even in this non-trivial spurious correlation setting: both the label smoothing and Mixup models have high test error for all settings while weight decay achieves a significantly lower test error for all $\lambda > 0$.

## 5 CONCLUSION

In this work, we have shown that label augmentation strategies such as label smoothing and Mixup exhibit a variance minimization effect (both in theory and in practice) that leads to lower variance intermediate representations and model outputs, which then correlate with better test performance on image classification benchmarks. A natural follow-up direction to our results is to investigate whether regularizers for encouraging lower variance features can be implemented directly to improve model performance, which would shed light on whether there is a causal relationship between this phenomenon and improved performance.

ETHICS STATEMENT

Although label smoothing and Mixup are used to train and fine-tune large-scale models, our results concerning them in this work have mostly been theoretical and explanatory. As a result, we do not anticipate any direct misuse of our results or any broader harmful impacts.

REPRODUCIBILITY STATEMENT

All proofs of the results in this paper can be found in Appendix A. The supplementary material contains the code necessary to generate all figures, with instructions on how to run each experiment.

ACKNOWLEDGMENTS

Rong Ge and Muthu Chidambaram were supported by NSF Award DMS-2031849 and CCF-1845171 (CAREER) during the completion of this work. Muthu would like to thank Annie Tang for thoughtful feedback during the early stages of this project.

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

## A    OMITTED PROOFS

### A.1    HELPER LEMMA

The following lemma lower bounds the Jensen gap in terms of the variance of the random variable being considered. We will make repeated use of it in the proofs of the label smoothing and Mixup results.

**Lemma A.1.** *Let $\phi : \mathbb{R}^d \to \mathbb{R}$ be a twice-differentiable convex function satisfying $\gamma_1 \mathrm{I}_d \prec \nabla^2 \phi \prec \gamma_2 \mathrm{I}_d$. Then for any square integrable random variable $X$ on $\mathbb{R}^d$ it follows that:*

$$\mathbb{E}[\phi(X)] - \phi(\mathbb{E}[X]) \in \left[ \frac{\gamma_1}{2} \mathrm{Var}(X), \frac{\gamma_2}{2} \mathrm{Var}(X) \right]. \tag{A.1}$$

*Proof.* From the assumption of the lemma it follows that $\phi(x) - \gamma_1 \|x\|^2 / 2$ is convex and $\phi(x) - \gamma_2 \|x\|^2 / 2$ is concave. Applying Jensen's inequality to each of these functions yields (A.1).  □

### A.2    PROOFS FOR SECTION 3.1

**Theorem 3.3.** *Let $w^*$ be the unique minimizer of $\ell_\beta(w)$ for $\beta > 0$ under $\pi$ satisfying Assumption 3.2. Then $\|w_\mathcal{H}^*\|^2 \geq \frac{1}{2}\|w^*\|^2$.*

*Proof.* Let us decompose the unique minimizer $w^*$ as $w^* = \alpha u^* + \beta v^*$, where $u^*$ and $v^*$ are orthonormal and satisfy $u_\mathcal{H}^* = 0, v_\mathcal{L}^* = 0$ (i.e. $u^*$ is a normalized version of the $\mathcal{L}$ components of $w^*$ and $v^*$ is the same but for the $\mathcal{H}$ components). We claim that $y\langle v^*, x \rangle > y\langle u^*, x \rangle$ for $\pi$-a.e. $(x, y)$. Indeed, if this were not the case then by the assumption on $\pi$ we could choose an orthonormal vector $z^*$ with $z_\mathcal{L}^* = 0$ that satisfies $y\langle z^*, x \rangle > y\langle u^*, x \rangle$ for $\pi$-a.e. $(x, y)$ and decrease the loss of $w^*$ by replacing $v^*$ with $z^*$.

Now suppose that $\alpha > \beta$. Then we claim that replacing $\alpha$ and $\beta$ with $\gamma = \sqrt{(\alpha^2 + \beta^2)/2}$ yields a solution with lower loss than $w^*$.

To see this, first observe that $2\gamma^2 = \alpha^2 + \beta^2$, so that the norm of the modified solution is the same as $w^*$. This implies that the weight decay penalty term in $\ell_\beta$ is unchanged.

Furthermore, we have that $\gamma \in (\beta, \alpha)$, and that $\gamma - \beta > \alpha - \gamma$. This follows from the fact that $\gamma \geq (\alpha + \beta)/2$ by construction. This, combined with the fact that $y\langle v^*, x \rangle > y\langle u^*, x \rangle$, then implies for $\pi$-a.e. $(x, y)$ that:

$$y\langle \gamma u^* + \gamma v^*, x \rangle > y\langle \alpha u^* + \beta v^*, x \rangle. \tag{A.2}$$

Which contradicts the minimality of $w^*$. Therefore, we must have $\alpha \leq \beta$, which gives the desired result.  □

**Corollary 3.4.** *If $w^*$ is the max-margin solution to $\ell_0(w)$, then the result of Theorem 3.3 still holds.*

*Proof.* We can apply the same decomposition as in the proof of Theorem 3.4; namely, $w^* = \alpha u^* + \beta v^*$ where $w^*$ is the max-margin solution and $u^*$ and $v^*$ are as before. Suppose again that $\alpha > \beta$ and let $\gamma = \sqrt{(\alpha^2 + \beta^2)/2}$. We claim that there exists $\epsilon \in (0, \gamma)$ such that $w = (\gamma - \epsilon)u^* + \gamma v^*$ satisfies $y\langle w, x, \geq \rangle 1$ for $\pi$-a.e. $(x, y)$, which would contradict $w^*$ being the max-margin solution since $\|w\|^2 < \|w^*\|^2$.

From the exact same logic as in the proof of Theorem 3.4, we have:

$$y\langle \gamma u^* + \gamma v^*, x \rangle - y\langle w^*, x \rangle > 0 \tag{A.3}$$

for $\pi$-a.e. $(x, y)$. Since $\mathrm{supp}(\pi)$ is compact, (A.3) attains a minimum $\kappa > 0$ over $\mathrm{supp}(\pi)$. Similarly, $y\langle u^*, x \rangle$ attains a finite maximum. By choosing $\epsilon$ such that $y\langle \epsilon u^*, x \rangle < \kappa$, we obtain the desired contradiction.  □

**Theorem 3.6.** *For $\alpha \in (0, 1)$ and $\pi$ satisfying Assumption 3.5, every minimizer $w^*$ of $\ell_{\mathrm{LS}, \alpha}(w)$ satisfies $\|w_\mathcal{H}^*\| < O(\|\Sigma_{YX, \mathcal{L}}\|)$.*

*Proof.* We follow the outline in the proof sketch. We first observe that there exists $M$ such that it suffices to consider $\|w\| \leq M$; this is due to the fact that for $\alpha > 0$, we have $\lim_{\|w\| \to \infty} |\ell_{\text{LS},\alpha}(w)| = \infty$. With this in mind, let us use $Z$ to denote $w^T Y X$ and use $\ell_{\text{LS},\alpha}(Z)$ to denote the loss in terms of this quantity. Then it follows that

$$\frac{\partial \ell_{\text{LS},\alpha}(Z)}{\partial Z} = \mathbb{E}\left[\sigma(Z) - 1 + \alpha/2\right], \tag{A.4}$$

$$\frac{\partial^2 \ell_{\text{LS},\alpha}(Z)}{\partial Z^2} = \mathbb{E}\left[\sigma(Z)(1 - \sigma(Z))\right], \tag{A.5}$$

where in both cases we applied the dominated convergence theorem, which is justified because $\ell_{\text{LS},\alpha}$ is smooth in $Z$ with bounded derivatives. Now since $\|w\| \leq M$ and the support of $X$ is compact, there exist $\gamma_1$ and $\gamma_2$ such that $\frac{\partial^2 \ell_{\text{LS},\alpha}(Z)}{\partial Z^2} \in (\gamma_1, \gamma_2)$, which implies that $\ell_{\text{LS},\alpha}$ is strongly convex in $Z$ and satisfies the conditions of Lemma A.1.

By Jensen's inequality, we then have that:

$$\ell_{\text{LS},\alpha}(Z) \geq \left(\frac{\alpha}{2} - 1\right) \log \sigma(\mathbb{E}[Z]) - \frac{\alpha}{2} \log \sigma(-\mathbb{E}[Z]). \tag{A.6}$$

Since $\mathbb{E}[Y X_{\mathcal{L}}] \neq 0$ by Assumption 3.5, we can choose $w$ such that $w_{\mathcal{H}} = 0$ and $\mathbb{E}[Z] = \sigma^{-1}(1 - \alpha/2)$, which minimizes the RHS of (A.6). Let us use OPT to denote this minimum. Then by Lemma A.1, we have for any $w$ chosen as described:

$$\ell_{\text{LS},\alpha}(w) - \text{OPT} \leq \frac{\gamma_2}{2} w^T \Sigma_{YX,\mathcal{L}} w. \tag{A.7}$$

On the other hand, if $w'$ is another solution satisfying $\|w'_{\mathcal{H}}\| > \epsilon$, then Lemma A.1 gives

$$\ell_{\text{LS},\alpha}(w') - \text{OPT} \geq \frac{\gamma_1 \rho \epsilon^2}{2}, \tag{A.8}$$

from which it is clear that for appropriate $\epsilon$ we cannot have $w'$ be a stationary point ($\rho$ above is the same as in Assumption 3.5). That we can take $\epsilon \to 0$ as $\|\Sigma_{YX,\mathcal{L}}\| \to 0$ follows from (A.7). $\qquad\square$

**Theorem 3.7.** *For any symmetric $\mathcal{D}_\lambda$ that is not a point mass on $0$ or $1$ and $\pi$ satisfying Assumption 3.5, every minimizer $w^*$ of $\ell_{\text{MIX},\mathcal{D}_\lambda}(w)$ satisfies $\|w^*_{\mathcal{H}}\| < O(\|\Sigma_{YX,\mathcal{L}}\|)$.*

*Proof.* Let us first outline the overall steps of the proof, and the differences with the label smoothing case.

1. We first show that the loss *conditioned on* $\lambda$ is strongly convex in $w^\top Z_\lambda$. The conditioning on $\lambda$ here is necessary because $\lambda$ is a random variable, unlike $\alpha$ in the label smoothing case. The overall goal here is to use the same argument as for label smoothing, i.e. show that we can achieve the optimal lower bound in terms of $\mathbb{E}[w^\top Z_\lambda]$ using only $w_{\mathcal{L}}$ and letting $\|\Sigma_{YX,\mathcal{L}}\| \to 0$.

2. We cannot explicitly minimize the conditional loss like we did with label smoothing, since it is not possible with a fixed choice of $w$ to achieve $\sigma^{-1}(\mathbb{E}[w^\top Z_\lambda]) = \lambda$ for every $\lambda$ simultaneously. Instead, we will show that a stationary point of the conditional loss exists that uses only the dimensions of $w_{\mathcal{L}}$.

3. Having shown the above, we can just reuse the same argument as before with Lemma A.1 to prove the desired result.

Let $\ell_{\text{MIX},\lambda}$ denote $\ell_{\text{MIX},\mathcal{D}_\lambda}$ with a fixed choice of $\lambda$ (i.e. after conditioning on $\lambda$), and let $R = w^\top Z_\lambda$. Then we can compute:

$$\frac{\partial \ell_{\text{MIX},\lambda}(R)}{\partial R} = \mathbb{E}\left[\lambda(\sigma(Y_1 R) - 1)Y_1 + (1 - \lambda)(\sigma(Y_2 R) - 1)Y_2\right], \tag{A.9}$$

$$\frac{\partial^2 \ell_{\text{MIX},\lambda}(R)}{\partial R^2} = \mathbb{E}\left[\lambda \sigma(Y_1 R)(1 - \sigma(Y_1 R)) + (1 - \lambda)\sigma(Y_2 R)(1 - \sigma(Y_2 R))\right], \tag{A.10}$$

where again we applied dominated convergence to the expectation with respect to $\pi$. Strong convexity follows from the same consideration as label smoothing; namely, we can consider $\|w\| \leq M$ as

$\ell_{\mathrm{MIX},\lambda}(R) \to \infty$ as $\|w\| \to \infty$ so long as $\lambda$ is not 0 or 1 and $\pi_Y(y) > 0$, and this consequently implies (A.10) is lower bounded by some positive real number.

Now by conditional Jensen's inequality, we obtain the following lower bound for $\ell_{\mathrm{MIX},\mathcal{D}_\lambda}$:

$$\ell_{\mathrm{MIX},\mathcal{D}_\lambda}(w) \geq \mathbb{E}_\lambda \left[ \mathbb{E}_{Y_1,Y_2} \left[ \ell_{\mathrm{MIX},\lambda}(\mathbb{E}\left[R \mid \lambda, Y_1, Y_2\right]) \right] \right]. \tag{A.11}$$

We now show that it is possible to minimize this lower bound while taking $w_{\mathcal{H}} = 0$. This is more difficult than it was in the label smoothing case, because it is no longer obvious that $\mathbb{E}[Y X_{\mathcal{L}}] \neq 0$ is sufficient for minimizing the RHS of (A.11) due to the expectation with respect to $\lambda$. However, we can show the existence of a stationary point with $w_{\mathcal{H}} = 0$, even though we cannot provide an explicit construction.

The idea is to consider the limiting behavior of (A.11) as we take the values of $w_{\mathcal{L}}$ to $-\infty$ and $\infty$. Note that we can take this limit into the expectation with respect to $\lambda$ by dominated convergence again. Let us consider the gradient with respect to $w$ of the RHS of (A.11). To make notation manageable, we will use $a_1 = \mathbb{E}[X \mid Y = 1], a_2 = \mathbb{E}[X \mid Y = -1]$, and $a_3 = \mathbb{E}[Z_\lambda \mid \lambda, Y_1 = 1, Y_2 = -1]$. We can then explicitly write out the gradient as the expectation with respect to $\lambda$ of the sum of the following three terms (considering the different cases for $Y_1, Y_2$):

$$\begin{aligned}
\boldsymbol{\nabla}_w \mathbb{E}_{Y_1,Y_2} \left[ \ell_{\mathrm{MIX},\lambda}(\mathbb{E}\left[R \mid \lambda, Y_1, Y_2\right]) \right] = {} & \pi_Y(1)^2 \left( \sigma(w^\top a_1) - 1 \right) a_1 \\
& - \pi_Y(-1)^2 \left( \sigma(-w^\top a_2) - 1 \right) a_2 \\
& + 2\pi_Y(1)\pi_Y(-1) \bigg( \lambda\big(\sigma(w^\top a_3) - 1\big) \\
& \qquad - (1 - \lambda)\big(\sigma(-w^\top a_3) - 1\big) \bigg) a_3.
\end{aligned} \tag{A.12}$$

The first two lines above are obtained from the fact that we can combine terms when $Y_1 = Y_2$, and the last line is by symmetry. Now we recall that by assumption $\mathbb{E}[Y X_{\mathcal{L}}] \neq 0$ and $\mathbb{E}[X] = 0$. Thus, WLOG, we can assume that $\mathbb{E}[X_{\mathcal{L}} \mid Y = 1]_i > 0$ and $\mathbb{E}[X_{\mathcal{L}} \mid Y = -1]_i < 0$ for each index $i$.

With this in mind, we consider first the case of what happens when the entries $w_{\mathcal{L}} \to \infty$. Since $w^\top a_1 > 0$ and $w^\top a_2 < 0$, the first two terms in (A.12) vanish (independent of $\lambda$). On the other hand, for the third term, there are two cases to consider. Depending on $\lambda$, we have that the entries of $a_3$ are either strictly negative or strictly positive, with the exceptional case of $\lambda = \pi_Y(1)$ in which $a_3 = 0$. If the entries of $a_3$ are strictly negative, then $\big(\sigma(-w^\top a_3) - 1\big) \to 0$ and the coefficient becomes negative, so the third term is positive. Similarly, if $a_3$ is strictly positive, the coefficient is positive and the third term is still positive. Thus, as $w_{\mathcal{L}} \to \infty$ every entry of (A.12) is positive.

Similar arguments show that the opposite is true when we take $w_{\mathcal{L}} \to -\infty$. Now by continuity of the gradient, it immediately follows that there is some choice of $w$ with only $w_{\mathcal{L}}$ non-zero such that the expectation of (A.12) with respect to $\lambda$ can be made to be zero. Using this choice of $w$ allows us to obtain an $R$ that minimizes the RHS of (A.11).

Now we have basically arrived at the same stage as the end of the label smoothing proof. By taking $\|\Sigma_{YX,\mathcal{L}}\| \to 0$ we can get arbitrary concentration around this optimal $R$, and by the same logic as the label smoothing proof the result follows. $\qquad\square$

## A.3 Proofs for Section 3.2

**Proposition 3.9.** *For $\alpha > 0$ and any $g : \mathbb{R}^d \to \Delta^{k-1}$, letting $\mathrm{OPT}_{\mathrm{LS},\alpha}$ denote the minimum of $\ell_{\mathrm{LS},\alpha}$, we have for a universal constant $C > 0$:*

$$\ell_{\mathrm{LS},\alpha}(g) \geq \mathrm{OPT}_{\mathrm{LS},\alpha} + C \sum_{y=1}^k \pi_Y(y) \mathrm{Var}\left( \mathbb{E}[g(X) \mid Y = y] \right). \tag{3.1}$$

*Proof.* Let us first define:

$$\ell_{\mathrm{LS},\alpha,y}(g) = -\mathbb{E}\left[ (1 - \alpha) \log g^y(X) + \frac{\alpha}{k} \sum_{i=1}^k \log g^i(X) \mid Y = y \right]. \tag{A.13}$$

Then we can decompose $\ell_{\mathrm{LS},\alpha}(g)$ as follows:

$$\ell_{\mathrm{LS},\alpha}(g) = \sum_{y=1}^{k} \pi_Y(y)\ell_{\mathrm{LS},\alpha,y}(g). \tag{A.14}$$

Once again, since we can restrict our attention to $g(X) \in [\gamma, 1-\gamma]$ for some $\gamma$ (as the loss goes to infinity if $g(X) = 1$ on a set of positive $\pi_X$-measure with $\alpha > 0$), it is easy to verify that (A.13) is strongly convex in $g(X)$. The desired result then follows by applying Lemma A.1 with the regular conditional distribution $\pi_{X|Y=y}$ for each term in (A.14). $\square$

**Proposition 3.10.** *For any $\mathcal{D}_\lambda$ that is not a point mass on $0$ or $1$ and any $g : \mathbb{R}^d \to \Delta^{k-1}$, letting* $\mathrm{OPT}_{\mathrm{MIX},\mathcal{D}_\lambda}$ *denote the minimum of $\ell_{\mathrm{MIX},\mathcal{D}_\lambda}$, we have for a universal constant $C > 0$ that:*

$$\ell_{\mathrm{MIX},\mathcal{D}_\lambda}(g) \geq \mathrm{OPT}_{\mathrm{MIX},\mathcal{D}_\lambda} + C \sum_{y_1=1}^{k} \sum_{y_2=1}^{k} \pi_Y(y_1)\pi_Y(y_2)\mathbb{E}_\lambda\left[\mathrm{Var}\left(\mathbb{E}[g(Z_\lambda) \mid y_1, y_2, \lambda]\right)\right]. \tag{3.2}$$

*Proof.* The proof follows an identical structure to that of Proposition 3.9. In particular, we again define the following conditional loss:

$$\ell_{\mathrm{MIX},\lambda,y_1,y_2} = -\mathbb{E}[\lambda \log g^{y_1}(Z_\lambda) + (1-\lambda) \log g^{y_2}(Z_\lambda) \mid y_1, y_2, \lambda]. \tag{A.15}$$

And we can then decompose $\ell_{\mathrm{MIX},\mathcal{D}_\lambda}$ as:

$$\ell_{\mathrm{MIX},\lambda}(g) = \sum_{y_1=1}^{k} \sum_{y_2=1}^{k} \pi_Y(y_1)\pi_Y(y_2)\mathbb{E}_{\lambda \sim \mathcal{D}_\lambda}[\ell_{\mathrm{MIX},\lambda,y_1,y_2}(g)]. \tag{A.16}$$

Since $\mathcal{D}_\lambda$ is not a point mass on $0$ or $1$, we can restrict ourselves to $g(X) \in [\gamma, 1-\gamma]$ for some $\gamma$ as before, and strong convexity in $g(X)$ of the conditional loss (A.15) again follows. We then apply Lemma A.1 to obtain the result. $\square$

## B  COMPARISON TO SETTINGS OF PRIOR WORK

Here we review the settings of Chidambaram et al. (2023) and Zou et al. (2023) in greater detail to provide a more precise comparison to the setting in our work.

**Setting of Chidambaram et al. (2023).** The authors consider a *multi-view* data model inspired by Allen-Zhu & Li (2021); namely, their data distribution $\pi$ is a multi-class distribution supported on $\mathbb{R}^{Pd} \times [k]$ where $P$ corresponds to the number of *patches* in each data point. Essentially, for $(x, y) \sim \pi$, we view $x$ as $x = \{x^{(1)}, x^{(2)}, ..., x^{(P)}\}$ with each $x^{(i)} \in \mathbb{R}^d$. The purpose of this partitioning is so that features related to the target class can appear in some patches and noise can appear in the remaining patches. In particular, the authors consider two target features per class ($v_{y,1}$ and $v_{y,2}$) that appear in a constant number of *signal patches*, while all other patches in a data point $x$ correspond to low magnitude *feature noise*, i.e. these patches consist of some linear combination of features $v_{s,j}$ for $s \neq y$. Furthermore, each signal patch has only a single feature (either $v_{y,1}$ or $v_{y,2}$) with a random weight $\beta$ such that for any signal patch $x^{(p)} = \beta v_{y,1}$ there is another patch $x^{(q)} = (C - \beta)v_{y,2}$ for a fixed parameter $C$. The authors emphasize that a model that has the same correlation with both $v_{y,1}$ and $v_{y,2}$ will achieve lower variance, as it will have a constant total correlation and be insensitive to the variation in $\beta$.

For this type of data distribution, the authors consider the training dynamics of a two-layer convolutional neural network with smoothed ReLU activations and non-trainable second layer weights (not an issue in this case, since second layer weights can be absorbed into the first layer). They analyze training using the empirical cross-entropy, as well as the Mixup cross-entropy for the specific case of a mixing distribution $\mathcal{D}_\lambda$ that is just a point mass on $1/2$ (Midpoint Mixup). The authors prove that, running gradient descent on the empirical cross-entropy leads (with high probability) to a model that only learns one feature for almost all classes, while doing the same for Midpoint Mixup yields a model that learns both features per class. The results are asymptotic; the authors consider all hyperparameters in their setup to be sufficiently large (even the number of classes $k$) or small.

**Setting of Zou et al. (2023).** Similar to Chidambaram et al. (2023), Zou et al. (2023) also works in a setting motivated by Allen-Zhu & Li (2021) and consider data consisting of patches. However, they focus on binary classification, and their data distribution $\pi$ is supported on $\mathbb{R}^{Pd} \times \{1, 2\}$. The data $x$ has exactly one (randomly selected) patch $x^{(i)}$ that contains a target feature; Zou et al. (2023) delineate one type of target feature $v$ as *common* and another type $v'$ as *rare*. There are also up to $b$ other patches in $x$ that consist of common features from other classes (i.e. feature noise, like in Chidambaram et al. (2023)). Lastly, different from Chidambaram et al. (2023), Zou et al. (2023) consider the leftover patches to consist of i.i.d. Gaussian noise.

Zou et al. (2023) also consider the training dynamics of a two-layer convolutional neural network with frozen second layer weights on this type of data distribution, but use a squared activation instead of a smoothed ReLU. Unlike the results of Chidambaram et al. (2023) which are stated entirely in terms of whether the features $v_{y,1}$ and $v_{y,2}$ are learned in a certain sense, Zou et al. (2023) directly prove a lower bound on the test error of models trained using gradient descent on the empirical cross-entropy while also showing that the test error of models trained on the empirical Mixup cross-entropy is vanishing small at some time step during model training. Essentially, this is due to the non-Mixup models failing to learn the rare feature $v'$ for both classes. Their results apply to Mixup with a mixing distribution $\mathcal{D}_\lambda$ that is any point mass on $(0.5, 1)$.

**Differences in our setting.** Both Chidambaram et al. (2023) and Zou et al. (2023) prove results concerning gradient descent dynamics, whereas our results directly consider the minimizers of the population losses associated with weight decay, label smoothing, and Mixup. Here our choice to work with the population losses is not substantially different from Chidambaram et al. (2023) and Zou et al. (2023), since although they work with the empirical losses they are in an asymptotic setting and large swaths of the proofs in both papers rely on concentration of measure arguments. However, we do differ substantially in that our main results apply to linear models – this makes our results less practical but technically much simpler than the results of Chidambaram et al. (2023) and Zou et al. (2023), with the added benefit that we can also handle any symmetric mixing distribution $\mathcal{D}_\lambda$.

Furthermore, since our main results work in this linear setting, we also adopt a much simpler data distribution setup. We do not consider our input data $x$ as partitioned into patches, instead we merely designate some subset of the input dimensions as "low variance" and the complementary subset as "high variance". In this sense, we do not have explicit feature vectors associated with each class $y$ like the $v_{y,1}, v_{y,2}$ of Chidambaram et al. (2023) or the $v, v'$ of Zou et al. (2023).

## C   ADDITIONAL EXPERIMENTS

### C.1   VISUALIZATION OF DECISION BOUNDARIES

To provide some intuition for Theorems 3.3 to 3.7, we visualize the decision boundaries of trained logistic regression models on 2-D data. In particular, denoting the classes as usual by $y \in \{-1, +1\}$, we consider a simple data distribution in which the first coordinate is distributed uniformly on $[y, 10y]$ and the second coordinate is fixed to be $0.1y$.

This data distribution is linearly separable in each coordinate; however, the second coordinate is fixed and thus has no conditional variance. Consequently, Theorems 3.6 and 3.7 predict that minimizing the population label smoothing and Mixup losses on this data should lead to learning a model whose decision boundary is aligned with the $x$-axis (i.e. we only use the second coordinate to determine which class we predict).

To verify this, we visualize the decision boundaries of logistic regression models trained on 500 points sampled from this distribution using multiple different settings of weight decay, label smoothing, and Mixup in Figure 5. We train for 500 epochs using full batch SGD with a learning rate of $10^{-2}$, although our results were not sensitive to these choices. We use a mixing distribution of $\text{Beta}(\alpha, \alpha)$ for Mixup (as is standard), and consider the canonical hyperparameter choices of $5 \times 10^{-4}$ for weight decay, 0.1 for label smoothing, and $\text{Beta}(1, 1)$ for Mixup (Zhang et al., 2018; Müller et al., 2020) in Figure 5 (a) and also check the effect of scaling these hyperparameters in Figure 5 (b) and (c).

As can be seen from the results, the label smoothing and Mixup decision boundaries are much closer to being aligned with the $x$-axis than the weight decay boundary, with the alignment getting stronger with more extreme choices of the hyperparameters. The fact that the boundaries are not exactly aligned with the $x$-axis can be attributed to the fact that we are not exactly minimizing the population

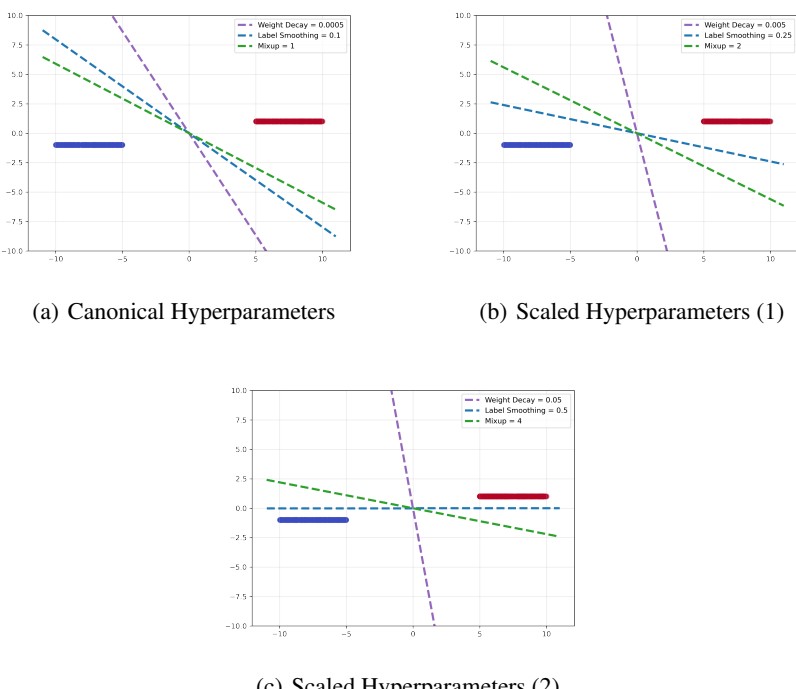

(a) Canonical Hyperparameters

(b) Scaled Hyperparameters (1)

(c) Scaled Hyperparameters (2)

Figure 5: Visualization of weight decay, label smoothing, and Mixup decision boundaries. Figure (a) considers canonical hyperparameter choices, and Figures (b) and (c) illustrate the effects of scaling these choices.

loss, since we are in the finite sample, fixed training horizon setting. That the label smoothing boundary is more aligned to the $x$-axis than the Mixup boundary is also to be expected, since the Mixup loss introduces randomness in the form of the mixing distribution which makes it more difficult to minimize. Lastly, the fact that the boundaries for both label smoothing and Mixup become more aligned with the $x$-axis at more extreme hyperparameter values can be intuitively explained by the fact that the loss incurred by predicting a probability close to 1 for either class increases as we scale the hyperparameters, i.e. we suffer more loss from relying on the high variance first coordinate.

*Remark* C.1. We should point out that, while our theoretical and empirical results in Figure 5 show a clear difference in the types of solutions learned when training linear classifiers using label smoothing, Mixup, and weight decay, it is not always the case that the augmented losses lead to different solutions than ERM (with possibly weight decay). Indeed, Chidambaram et al. (2021) and Oh & Yun (2023) both showed that for different settings of Gaussian data, ERM and Mixup can lead to learning the same solution. However, the settings of these prior works don't fall within our scope as we consider distributions with *compact support*, which ends up being an important property for proving our results.

## C.2 DIRECT VERIFICATION OF THEORY

Here we directly analyze the training of logistic regression models on a synthetic data distribution that exactly follows the assumptions of Definition 3.1; the following definition generalizes the 2-D distribution we visualized in Section C.1.

**Definition C.2** (Synthetic Data). We define a distribution $\pi_\gamma$ parameterized by $\gamma$ (where $0 < \gamma < 1$) on $\mathbb{R}^d \times \{-1, 1\}$ with $\pi_{\gamma,Y}(1) = 1/2$ and $x \sim \pi_{\gamma,X|Y=y}$ satisfying:

1. **First $\lfloor d/2 \rfloor$ dimensions are constant but small.** We have $x_i = \gamma y$ for $i \leq \lfloor d/2 \rfloor$.

2. **Last $d - \lfloor d/2 \rfloor$ dimensions are high variance.** We have $x_i \sim \text{Uniform}([y, 100y])$ for $i > \lfloor d/2 \rfloor$.

In other words, we consider data where (conditional on the label) the first half of the dimensions are fixed (corresponding to $\mathcal{L}$) and the second half are i.i.d. high variance uniform (corresponding to $\mathcal{H}$). We sample $n = 5000$ data points according to Definition C.2 with $d = 10$ and $\gamma = 0.1$ and train logistic regression models across a range of settings for weight decay, label smoothing, and Mixup. The choice of $\gamma$ here, as well as the range of values for the dimensions in $\mathcal{H}$, is relatively arbitrary; we verified our empirical results hold for different scales of $\gamma$ and ranges for $\mathcal{H}$. The empirical results also do not depend on the fact that the $\mathcal{L}$ coordinates are zero variance – we checked that they still hold when adding a small magnitude uniform noise to the $\mathcal{L}$ coordinates.

For model training, we consider 20 uniformly spaced values in $[0, 0.1]$ for the weight decay $\lambda$ parameter and in $[0, 0.75]$ for the label smoothing $\alpha$ parameter, where the upper bound for the label smoothing parameter space is obtained from the experiments of Müller et al. (2020). For Mixup, we fix the mixing distribution to be the canonical choice of $\mathrm{Beta}(\alpha, \alpha)$ introduced by Zhang et al. (2018) and consider 20 uniformly spaced $\alpha$ values in $[0, 8]$ (with $\alpha = 0$ corresponding to ERM), which effectively covers the range of Mixup hyperparameter values used in practice.

We train all models for 100 epochs using AdamW with the standard hyperparameters of $\beta_1 = 0.9, \beta_2 = 0.999$, a learning rate of $5 \times 10^{-3}$, and a batch size of 500. At the end of training, we compute $\|w_{\mathcal{H}}\|$ (i.e. the norm of the weight vector in the last 5 dimensions) for each trained model. For each model setting, we report the mean and standard deviation of $\|w_{\mathcal{H}}\|$ over 5 training runs in Figure 6.

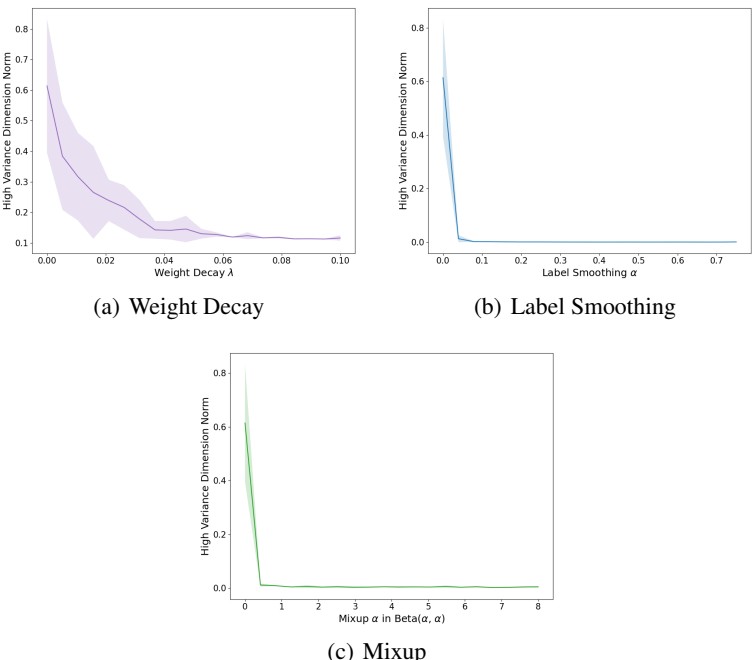

(a) Weight Decay        (b) Label Smoothing

(c) Mixup

Figure 6: Final weight norm of logistic regression model in the high variance dimensions ($\|w_{\mathcal{H}}\|$) for various hyperparameter settings on the synthetic data distribution of Section C.2.

As can be seen from the results, the weight decay models always have non-trivial values for $\|w_{\mathcal{H}}\|$ (even for large values of $\lambda$), whereas the label smoothing and Mixup models very quickly converge to a $\|w_{\mathcal{H}}\|$ of effectively zero as their respective hyperparameters move away from the ERM regime ($\alpha = 0$ in both cases). This matches the behavior predicted by Theorems 3.3 to 3.7.

## C.3 FULL RESNET RESULTS

Here we collect final test error/variance results for the plots shown in Section 4.1 and also provide analogous plots and results for ResNet-50 and ResNet-101 models. In all of the following, we abbreviate weight decay as "WD" and label smoothing as "LS".

### C.3.1 RESNET-18 FINAL RESULTS

Tables 1 and 2 show the end-of-training test error, activation variance, and output variance results for the experiments in Figure 1.

| Method | Test Error | Activation Variance | Output Variance |
|---|---|---|---|
| ERM + WD | $8.40 \pm 0.58$ | $198 \pm 5$ | $0.133 \pm 0.008$ |
| ERM + WD + LS | $7.96 \pm 0.30$ | $33.0 \pm 5.9$ | $0.099 \pm 0.003$ |
| ERM + WD + Mixup | $\mathbf{5.90} \pm 0.21$ | $18.4 \pm 0.4$ | $0.059 \pm 0.002$ |

Table 1: Final results (mean test error/variance and one standard deviation over 5 runs) for ResNet-18 experiments on CIFAR-10.

| Method | Test Error | Activation Variance | Output Variance |
|---|---|---|---|
| ERM + WD | $31.59 \pm 0.36$ | $673 \pm 23$ | $0.406 \pm 0.007$ |
| ERM + WD + LS | $28.55 \pm 4.7$ | $54.7 \pm 5.6$ | $0.188 \pm 0.027$ |
| ERM + WD + Mixup | $\mathbf{26.83} \pm 0.44$ | $82.2 \pm 1.2$ | $0.143 \pm 0.004$ |

Table 2: Final results (mean test error/variance and one standard deviation over 5 runs) for ResNet-18 experiments on CIFAR-100.

### C.3.2 RESNET-50 ALL RESULTS

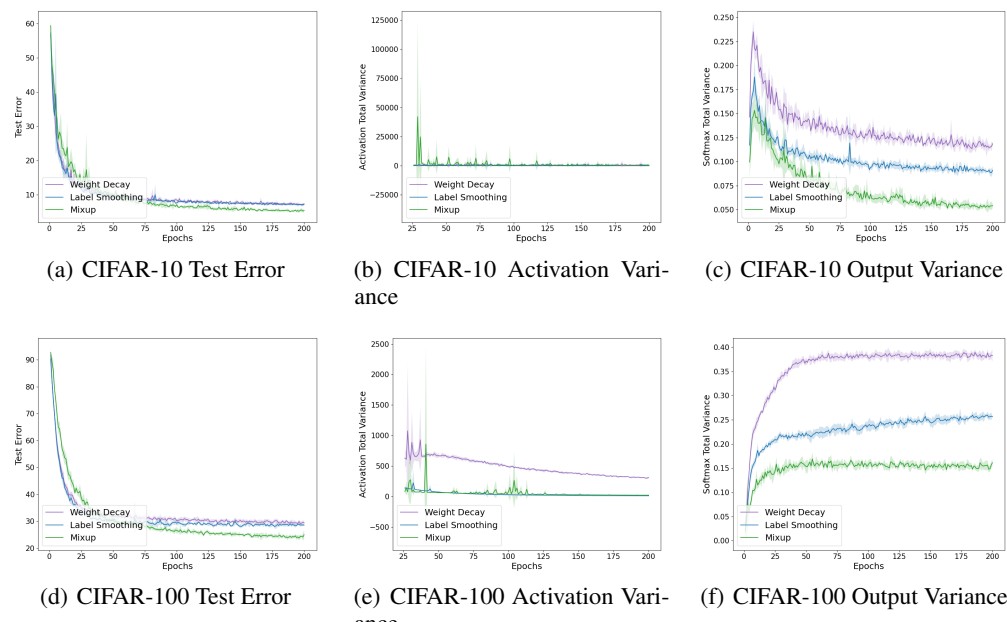

(a) CIFAR-10 Test Error

(b) CIFAR-10 Activation Variance

(c) CIFAR-10 Output Variance

(d) CIFAR-100 Test Error

(e) CIFAR-100 Activation Variance

(f) CIFAR-100 Output Variance

Figure 7: ResNet-50 final test errors, penultimate layer activation variances, and output probability variances on CIFAR-10 and CIFAR-100.

We train ResNet-50 models under the same hyperparameters as in Section 4.1, except for a batch size of 512 due to memory constraints. ResNet-50 results analogous to those shown in Figure 1 are shown in Figure 7. Similarly, final error and variance results are shown in Tables 3 and 4.

We observe that while the same trends hold as in the case of ResNet-18, there is significantly more variance in the computed activation variances for Mixup on CIFAR-10. In this particular case, we found that there were still significant oscillations in activation variances even towards the end of the training horizon. This may in part be attributable to reduced batch size, but we did not investigate this further.

| Method | Test Error | Activation Variance | Output Variance |
|---|---|---|---|
| ERM + WD | $7.36 \pm 0.13$ | $551 \pm 711$ | $0.119 \pm 0.002$ |
| ERM + WD + LS | $7.24 \pm 0.26$ | $6.35 \pm 3.47$ | $0.091 \pm 0.004$ |
| ERM + WD + Mixup | $\mathbf{5.41} \pm 0.37$ | $95 \pm 115$ | $0.054 \pm 0.003$ |

Table 3: Final results (mean test error/variance and one standard deviation over 5 runs) for ResNet-50 experiments on CIFAR-10.

| Method | Test Error | Activation Variance | Output Variance |
|---|---|---|---|
| ERM + WD | $29.41 \pm 0.94$ | $310 \pm 14$ | $0.383 \pm 0.012$ |
| ERM + WD + LS | $28.58 \pm 0.28$ | $13.14 \pm 0.20$ | $0.257 \pm 0.001$ |
| ERM + WD + Mixup | $\mathbf{24.94} \pm 1.63$ | $22.95 \pm 1.52$ | $0.161 \pm 0.007$ |

Table 4: Final results (mean test error/variance and one standard deviation over 5 runs) for ResNet-50 experiments on CIFAR-100.

### C.3.3 RESNET-101 ALL RESULTS

We also train ResNet-101 models under the same hyperparameters as in Section 4.1, except once again for a batch size of 512 due to memory constraints. ResNet-101 results analogous to those shown in Figure 1 are shown in Figure 8. Similarly, final error and variance results are shown in Tables 5 and 6.

Here we see that the variance oscillation behavior that we mentioned in Appendix C.3.2 is even more pronounced for the CIFAR-10 results, suggesting that this behavior is amplified for larger models. Once again, it is not clear what properties of CIFAR-10 lead to highly oscillatory activation variances for some initializations, but we again suspect that reduced batch size in training at least plays some role. The CIFAR-100 results remain consistent, although there is still some oscillatory behavior for the Mixup results.

| Method | Test Error | Activation Variance | Output Variance |
|---|---|---|---|
| ERM + WD | $6.87 \pm 0.17$ | $491 \pm 235$ | $0.110 \pm 0.002$ |
| ERM + WD + LS | $6.90 \pm 0.26$ | $2280 \pm 2292$ | $0.088 \pm 0.003$ |
| ERM + WD + Mixup | $\mathbf{5.23} \pm 0.18$ | $1701 \pm 1586$ | $0.054 \pm 0.002$ |

Table 5: Final results (mean test error/variance and one standard deviation over 5 runs) for ResNet-101 experiments on CIFAR-10.

| Method | Test Error | Activation Variance | Output Variance |
|---|---|---|---|
| ERM + WD | $28.69 \pm 0.73$ | $283 \pm 18$ | $0.373 \pm 0.008$ |
| ERM + WD + LS | $27.74 \pm 0.55$ | $12.01 \pm 0.23$ | $0.269 \pm 0.005$ |
| ERM + WD + Mixup | $\mathbf{23.55} \pm 0.98$ | $115 \pm 136$ | $0.157 \pm 0.01$ |

Table 6: Final results (mean test error/variance and one standard deviation over 5 runs) for ResNet-101 experiments on CIFAR-100.

### C.4 OUTPUT PROBABILITY VARIANCE ANALYSIS

A natural explanation for why output probability variance decreases for label smoothing and Mixup in Figures 1, 7, and 8 is that label smoothing and Mixup improve test error and consequently have less variability in their outputs due to fewer mistakes. Firstly, looking carefully at the label smoothing results in the previous subsections shows this cannot be the full cause, as in Table 5 label smoothing leads to worse test error than the baseline of ERM + WD but still leads to lower output probability variance.

In Tables 7 to 9, we compute the average output probability variance for each class but consider only the probability associated with the target class (i.e. for all points with label $y$, we compute the variance of the predicted probabilities corresponding to $y$). In all cases, the target output variance alone leaves a significant fraction of the overall output variance unexplained.

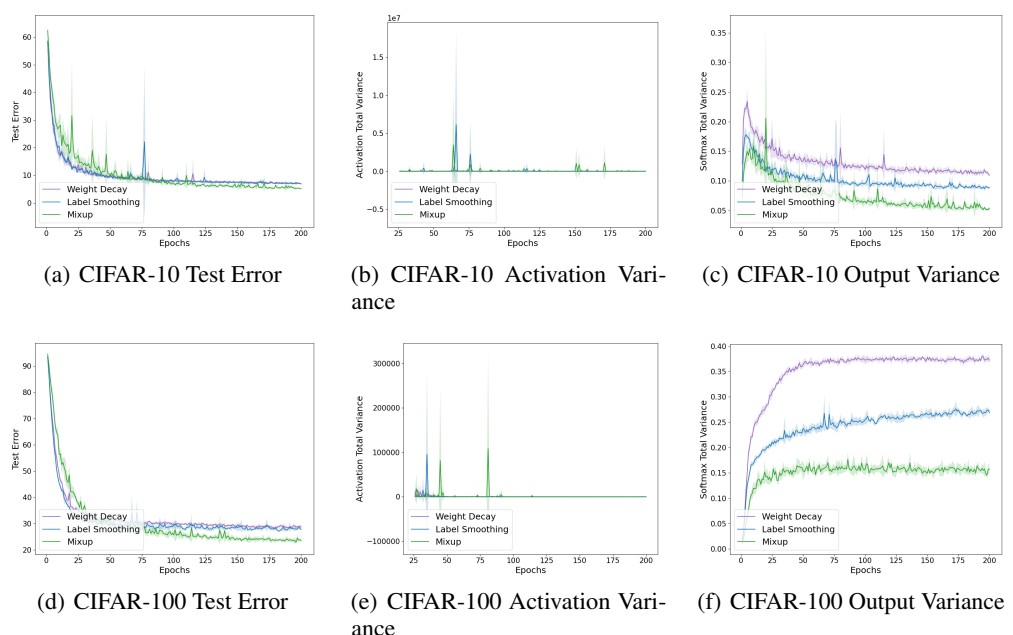

(a) CIFAR-10 Test Error  (b) CIFAR-10 Activation Variance  (c) CIFAR-10 Output Variance

(d) CIFAR-100 Test Error  (e) CIFAR-100 Activation Variance  (f) CIFAR-100 Output Variance

Figure 8: ResNet-101 final test errors, penultimate layer activation variances, and output probability variances on CIFAR-10 and CIFAR-100.

| Method | Target Output Variance (CIFAR-10) | Target Output Variance (CIFAR-100) |
|---|---|---|
| ERM + WD | $0.066 \pm 0.004$ | $0.172 \pm 0.002$ |
| ERM + WD + LS | $0.050 \pm 0.002$ | $0.119 \pm 0.002$ |
| ERM + WD + Mixup | $0.029 \pm 0.002$ | $0.087 \pm 0.002$ |

Table 7: Target output probability variance for ResNet-18.

| Method | Target Output Variance (CIFAR-10) | Target Output Variance (CIFAR-100) |
|---|---|---|
| ERM + WD | $0.059 \pm 0.001$ | $0.166 \pm 0.004$ |
| ERM + WD + LS | $0.046 \pm 0.002$ | $0.131 \pm 0.001$ |
| ERM + WD + Mixup | $0.027 \pm 0.002$ | $0.093 \pm 0.004$ |

Table 8: Target output probability variance for ResNet-50.

| Method | Target Output Variance (CIFAR-10) | Target Output Variance (CIFAR-100) |
|---|---|---|
| ERM + WD | $0.054 \pm 0.001$ | $0.162 \pm 0.002$ |
| ERM + WD + LS | $0.044 \pm 0.001$ | $0.131 \pm 0.002$ |
| ERM + WD + Mixup | $0.026 \pm 0.002$ | $0.088 \pm 0.004$ |

Table 9: Target output probability variance for ResNet-101.

