# OpenReview forum: "For Better or For Worse? Learning Minimum Variance Features With Label Augmentation"
_ICLR.cc/2025/Conference — ICLR 2025 Poster_

### Official Review · Reviewer_tHC2 · 2024-10-27

**Soundness:** 3
**Presentation:** 2
**Contribution:** 2
**Rating:** 5
**Confidence:** 2

**Summary:**

In this research, the authors investigate label augmentation methods such as label smoothing and Mixup within deep learning frameworks. These techniques adjust both the input data and labels to enhance training outcomes. The study delves into how these methods impact feature learning and reduce output variance. The findings indicate that label augmentation helps models develop low-variance features, improving classification results on some benchmarks, though it may increase susceptibility to misleading data patterns. With theoretical backing and empirical results, the paper highlights the strengths and drawbacks of label smoothing and Mixup. The authors conclude that while these approaches bolster model robustness, they may inadvertently promote reliance on low-variance features, affecting model generalization.

**Strengths:**

1. This paper rigorously analyzes label smoothing and Mixup with mathematical proofs, building a solid theoretical foundation.

**Weaknesses:**

1. Overall, the authors' argument that both Mixup and label smoothing are effective in reducing variance is not particularly surprising and has been a well-known perspective for some time. For this reason, Mixup has been used for calibration due to its generalizability to the underlying distribution of training datasets.

2. The authors should conduct experiments with diverse spurious correlation setups. For example, both label smoothing and Mixup are widely used to mitigate noisy label learning, which is one common spurious correlation setup. I believe these previous works contradict the authors' arguments.

[1] When and How Mixup Improves Calibration, ICML 2022 \
[2] How Does Mixup Help with Robustness and Generalization?, ICLR 2021 \
[3] Does Label Smoothing Mitigate Label Noise?, ICML 2020 \
[4] mixup: BEYOND EMPIRICAL RISK MINIMIZATION, ICLR 2018

**Questions:**

Is there any theoretical investigation into whether Mixup and label smoothing perform worse in situations involving spurious correlations? I think the limited experimental setups are insufficient to support a counter-intuitive argument compared to the extensive body of previous work on model calibration and noisy label learning.

---

> ### Author Response · Authors · 2024-11-18
> **Response to Reviewer tHC2**
>
> We would like to thank Reviewer tHC2 for taking the time to review our paper, and we hope to address the stated weaknesses below.
>
> ## Weaknesses
>
> 1. While we agree that prior works have definitely pointed towards a variance minimization effect for Mixup and label smoothing (e.g. in relation to neural collapse), we are not aware of any work that theoretically establishes this from a feature learning perspective. The goal of our work was to unify and simplify some of the ideas in previous feature learning analyses of Mixup [1, 2].
>
> 2. The findings of prior works do not contradict our results -- we now further emphasize in the revisions that low variance features can either be the features that generalize well or the ones that are spuriously correlated with the targets. We hypothesize that due to the repeatedly reported success of label smoothing and Mixup, in practical settings it is likely the case that there are low variance latent features that generalize well as opposed to being spuriously correlated with the targets. We have included multiple different spurious correlation setups in Section 4.2 of our paper (and now also Appendix C.2); we feel that a complete investigation of the existence of low variance/high variance latent features in standard benchmarks is outside the scope of our current work.
>
> ## Questions
>
> 1. Other than the theoretical results we prove, which can be translated to show poor generalization in settings where the spurious correlations are low variance (like in Section 4.2), we are not aware of other works that try to make this point.
>
> We hope the above provides further clarity regarding our paper and we are happy to answer any further questions you may have.
>
> [1] Chidambaram, M., Wang, X., Wu, C., & Ge, R. (2022). Provably Learning Diverse Features in Multi-View Data with Midpoint Mixup. International Conference on Machine Learning.
>
> [2] Zou, D., Cao, Y., Li, Y., & Gu, Q. (2023). The Benefits of Mixup for Feature Learning. International Conference on Machine Learning.

---

> > ### Comment · Reviewer_tHC2 · 2024-11-21
> >
> > Thanks for the responses corresponding, most of my concerns are resolved, however, I still worried about the experiments on single type of spurious correlations. Thus, I will raised my rating just slightly (3 $\rightarrow$ 5).

---

### Official Review · Reviewer_LPKu · 2024-10-30

**Soundness:** 3
**Presentation:** 2
**Contribution:** 3
**Rating:** 6
**Confidence:** 3

**Summary:**

The paper studies feature learning in training with label augmentation techniques like Mixup and label smoothing, etc.. Using a linear binary classification model, and proper conditions on the data distribution, the authors have theoretically shown that in ERM training, the learned model tends to assign more correlations on the high-variance features, while in label augmentation training, it's quite the opposite. They then empirically show that such a behavior can lead to both positive and negative consequences. In particular, they show that the model's learning low variance features correlates to better generalization, while the model's learning only the low variance features, if happening, can  harm the generalization.

**Strengths:**

1. The studying of feature learning problem on label smoothing is novel
2. The paper have uncovered some interesting and new findings about the phenomenons and properties existed in label augmentation techniques like Mixup and label smoothing
3. The theoretical contributions and their proofs are valid
4. The experiments are comprehensive, and they show not only the validity of the theoretical results, but also the correlationship between those theoretical results and the impact of the label augmentation techniques on model performance

**Weaknesses:**

Please refer to the Questions session for details of the weaknesses listed here

1. While the concept of "high variancce features" and "low variance feature" appear to be the essential components of the theoretical content, the whole paper seems to lack a strict definition of them. Perhaps that's a common knowledge in the field of feature learning, but since this paper is mostly theory-based, it's better to provide more mathematically strict definition of these concepts.
2. The logic connection from section 3.1 to section 3.2 is not so clear, in fact, from my understanding, they are talking about different things.
3. Since the main theoretical  results are built based on a simple model, one would love to see first a experiment done on such a toy model to show the validity of the theoretical results in the most direct way.
4. Some notations and terminologies can be perfected
5. There lacks an intuitive explanation or an insight behind the connection between learning of low variance features and generalization performance of the trained model.

**Questions:**

1. Line 135 on page 3. The identity matrix shouldd be denoted by mathematical expression "$I_d$" rather than text "Id" right?
2. In the context of binary classification, the label variable $y$ should be $Y\in \{-1, +1\}$ right? But in that case, the cross-entropy loss defined in equation 2.1 cannot be applied on binary classification. Or, is the paper implicitly regarding the term "$k$-class classification" to be distinct from the term "binary classification"?
3. In equation 2.4, I think the second term inside the expectation should be "$\dfrac{\alpha}{k-1}\sum_{i\in [k]/{Y}}\log{g^i(X)}$".
4. In the definition of Mixup loss in equation 2.8, are the points pairs $(X_1,Y_1)$ and $(X_2,Y_2)$ sampled independently from $\pi$? In that context, I would denote it as $(X_{1:2},Y_{1:2})\overset{\mathrm{iid}}{\sim}\pi$ or something like $(X_{1:2},Y_{1:2})\sim\pi\bigotimes\pi$
5. What are the mathematical definitions of low variance features and high variance features? Is it the overall variance? Or in-class variances? Also, how would one justify if a feature is high variance or low? Should there be a predefined threshold? Or maybe even a post-defined threshold, meaning threshold is defined after the groups of high variance feature and low variance features are manually partitioned?
6. Before assumption 3.2, $\mathcal{L}$ and $\mathcal{H}$ are at first defined as subsets of $[d]$, and then regarded as sets of vectors. I understand this is only for the sake of simplicities in notations, but I would suggest that the paper simply uses different notations.
7. In assumption 3.5, how does the first part ensure that we can obtain good solution using low variance dimensions? First, I think if a data is class-balanced (the marginal distribution on the classes is discrete uniform), and given an early assumption that $\mathbb{E}[X]=0$, then $\mathbb{E}[YX_\mathcal{L}]$ should always be 0. Second, if the data is imbalanced, then this first part might hold true, but then does it mean we are only dealing with class-imbalanced data from now on? Also, even if the data is imbalanced and $\mathbb{E}[YX_\mathcal{L}]$ equals 0, it may still be possible to find a good solution.
8. The assumptions on the data in the setting of linear binary classification don't seem to be so unrealistic. Can the authors provide some toy example experiments to verify the theoretical results more directly?
9. I was expecting section 3.2 would be a generalization of the results in section 3.1. But in section 3.1, the conditions are about the variances in the data dimension. But in section 3.2, somehow it's all about the variances in the model output. I am aware of the statement made in remark 3.11, but I still don't understand how does this section help us gain more understanding of the link between the model's behavior of learning low variance features and the change in the model's generalization peformance?
10. Line 368 on page 7. It better to write "$5\times 10^{-4}$" instead of "$*$".
11. Is there and inituitions or insight as of how does the variance of the learned features affect the performance of the model? Without this, it's pretty hard to use the work in this paper as a cornerstone to initiate further studies, such as the mechanism of label augmentation techniques (in other words, why do they work), the limitations and possible flaws of these techniques, how to improve these techniques or design new algorithms, etc.. This is why I feel like the contribution to the research area in this paper is only fair.

---

> ### Author Response · Authors · 2024-11-18
> **Response to Reviewer LPKu (Part 1)**
>
> We would like to thank Reviewer LPKu for taking the time to review our paper and for finding our results to be novel, and we appreciate the many detailed comments which we hope to address below.
>
> ## Weaknesses/Questions
>
> > Line 135 on page 3. The identity matrix shouldd be denoted by mathematical expression "$\mathrm{I}_d$" rather than text "Id" right?
>
> While we consider both notations to be standard, we have changed to $\mathrm{I}_d$ to better emphasize the dimension and also pointed this out in the notation.
>
> > In the context of binary classification, the label variable should be $Y \in \{-1, +1\}$ right? But in that case, the cross-entropy loss defined in equation 2.1 cannot be applied on binary classification. Or, is the paper implicitly regarding the term "$k$-class classification" to be distinct from the term "binary classification"?
>
> We are considering $k$-class classification to be distinct from binary classification, since taking $y \in \{-1, +1\}$ for binary classification (as opposed to $y \in \{0, 1\}$) makes it much easier to state the binary classification results/experiments. We make this distinction in the discussion right after we introduce our notation in Section 2.
>
> > In equation 2.4, I think the second term inside the expectation should be "$\frac{1}{\alpha - 1} \sum_{i \in [k] \setminus Y} \log g^i(X)$".
>
> While there are different stated versions of the label smoothing loss in the literature (depending on the valid range of smoothing hyperparameter), we prefer our version when we consider $\alpha \in (0, 1)$ (and our results use this version as stated). By taking the summation $\frac{\alpha}{k}\sum_{i = 1}^k \log g^i(x)$ over all classes, we ensure that the probability mass assigned to the true target class never becomes smaller than that assigned to any other class, at the cost of interpreting $\alpha$ as the total mass assigned to all other classes.
>
> > In the definition of Mixup loss in equation 2.8, are the points pairs $(X_1, Y_1)$ and $(X_2, Y_2)$ sampled independently from $\pi$? In that context, I would denote it as $(X_{1:2}, Y_{1:2}) \stackrel{i.i.d.}{\sim} \pi$ or $(X_{1:2}, Y_{1:2})  \sim \pi \otimes \pi$.
>
> The pairs are indeed sampled independently, and we have fixed the notation as you have suggested to be $\pi \otimes \pi$.
>
> > What are the mathematical definitions of low variance features and high variance features? Is it the overall variance? Or in-class variances? Also, how would one justify if a feature is high variance or low? Should there be a predefined threshold? Or maybe even a post-defined threshold, meaning threshold is defined after the groups of high variance feature and low variance features are manually partitioned?
>
> Our notions of low and high variance are codified entirely in Assumption 3.5; i.e. it is a condition on the covariance matrix of the data multiplied by the label and restricted to certain dimensions. We have stated it this way for brevity, but you are definitely correct in that this condition could also be stated in terms of the class-conditional covariance matrices. In particular, "high variance" here just means that those dimensions have nontrivial variance no matter how you combine them linearly. In our experiments, we differentiate high variance and low variance by the individual variances of the dimensions of the input; the low variance "feature" corresponds to those dimensions of the input whose variance falls below some small threshold (which we can vary, so long as it is much smaller than the variance of the high variance dimensions).
>
> > Before assumption 3.2, $\mathcal{L}$ and $\mathcal{H}$ are at first defined as subsets of $[d]$, and then regarded as sets of vectors. I understand this is only for the sake of simplicities in notations, but I would suggest that the paper simply uses different notations.
>
> While we agree that it is regrettable that we abuse notation by saying $v \in \mathcal{L}$, we prefer this notation to repeatedly restating $v_i = 0$ for all $i \in \mathcal{L}$.

---

> > ### Author Response · Authors · 2024-11-18
> > **Response to Reviewer LPKu (Part 2)**
> >
> > > In assumption 3.5, how does the first part ensure that we can obtain good solution using low variance dimensions? First, I think if a data is class-balanced (the marginal distribution on the classes is discrete uniform), and given an early assumption that $\mathbb{E}[X] = 0$, then $\mathbb{E}[YX_{\mathcal{L}}]$ should always be 0. Second, if the data is imbalanced, then this first part might hold true, but then does it mean we are only dealing with class-imbalanced data from now on? Also, even if the data is imbalanced and $\mathbb{E}[YX_{\mathcal{L}}]$ equals 0, it may still be possible to find a good solution.
> >
> > We apologize for not making Assumption 3.5 clearer. In fact, we were implicitly considering the balanced class setting, since for $YX$ to concentrate on a point mass with $\mathbb{E}[X] = 0$ we need classes to be balanced (this is now clarified in a footnote). For example, considering $\mathbb{E}[X \mid Y = +1] = 1$ and $\mathbb{E}[X \mid Y = -1] = -1$ we have $\mathbb{E}[YX] = 1$. We could have alternatively stated our assumptions in terms of the norm of the class-conditional covariance matrices going to zero to allow for other distributions of classes.
> >
> > > The assumptions on the data in the setting of linear binary classification don't seem to be so unrealistic. Can the authors provide some toy example experiments to verify the theoretical results more directly?
> >
> > We now include an experimental verification directly based on the data distribution of our theoretical setting in Appendix C.2.
> >
> > > I was expecting section 3.2 would be a generalization of the results in section 3.1. But in section 3.1, the conditions are about the variances in the data dimension. But in section 3.2, somehow it's all about the variances in the model output. I am aware of the statement made in remark 3.11, but I still don't understand how does this section help us gain more understanding of the link between the model's behavior of learning low variance features and the change in the model's generalization peformance?
> >
> > In the case of Section 3.2, we focus on model outputs to be model agnostic, but the intuition is to think of some large nonlinear model which is essentially a linear layer operating on learned features. With a weight decay constraint, we don't necessarily need the model outputs to be low variance to minimize the loss since it might be the case that we don't want to learn low variance features from the data due to them having low norm as well, which would lead to large last layer weights and consequently a high $L^2$ penalty. On the other hand, if there exist some low variance representations in the data that do generalize well, then these will consequently lead to lower model output variances and be favored by Mixup and label smoothing.
> >
> > > Line 368 on page 7. It better to write "$5 \times 10^{-4}$" instead of "$*$".
> >
> > We have fixed the writing to be $5 \times 10^{-4}$.
> >
> > > Is there and inituitions or insight as of how does the variance of the learned features affect the performance of the model? Without this, it's pretty hard to use the work in this paper as a cornerstone to initiate further studies, such as the mechanism of label augmentation techniques (in other words, why do they work), the limitations and possible flaws of these techniques, how to improve these techniques or design new algorithms, etc.. This is why I feel like the contribution to the research area in this paper is only fair.
> >
> > To provide more intuition for how the variance of the features affect the behavior of the learned model, we have added Appendix C.1 that visualizes decision boundaries for weight decay, label smoothing, and Mixup. In practical settings, one intuition for why it might be good to learn low variance features for generalization is that the high variance features correspond to noise in the training data, which is a perspective partially analyzed by prior work [1, 2]. Our goal in this work was just to clarify that the underlying phenomenon in the prior works is strongly related to minimizing variance.
> >
> > We hope the above addresses the main pointed out weaknesses/questions regarding our work, and we are happy to answer any further questions that may arise.
> >
> > [1] Chidambaram, M., Wang, X., Wu, C., \& Ge, R. (2022). Provably Learning Diverse Features in Multi-View Data with Midpoint Mixup. International Conference on Machine Learning.
> >
> > [2] Zou, D., Cao, Y., Li, Y., \& Gu, Q. (2023). The Benefits of Mixup for Feature Learning. International Conference on Machine Learning.

---

> > > ### Comment · Reviewer_LPKu · 2024-11-18
> > >
> > > Rebuttal acknowledged, raised rating from 5 to 6

---

### Official Review · Reviewer_Gh1c · 2024-10-31

**Soundness:** 3
**Presentation:** 2
**Contribution:** 2
**Rating:** 6
**Confidence:** 3

**Summary:**

In this paper the authors provide theoretical results pertaining to the variance of features in relation to training with label augmentation (Label Smoothing and Mixup). They show that for a binary linear logistic classifier trained on a certain distribution where there is a high and a low variance feature, label augmentation leads optimisation to only consider the low variance feature. For general models and classification they provide a quantative lower bound on how the variance of output probabilities contributes to the loss. The authors then provide some experiments to empirically verify the theoretical results.

**Strengths:**

- The paper provides some new theoretical results that contribute to better understanding of the behaviour of label augmentation approaches. I appreciate work that tries to advance knowledge in deep learning rather than simply chasing benchmark scores so this is a big plus for me.
- The experiments are well-targeted to verify the theoretical results.

**Weaknesses:**

I would like to clarify that I did not have time to comb the fine details of the paper or verify the theory, however I believe I have allocated a reasonable amount of time to reading from start to end. I am also not a "theoretical" machine learning researcher personally, and I am more familiar with the literature for label smoothing than Mixup, which should help put my later comments in context. I welcome the authors to correct any mistakes/misunderstandings that I make in this review.

1. The presentation of the paper inhibits easy understanding of the theoretical takeaways for a general machine learning audience (i.e. the average researcher using label augmentation). There isn't a single visual aid to illustrate any of the theoretical results, making it both difficult to intuit the theory and understand the key takeaways. For example, for the linear binary classifier, the paper would be greatly aided with a simple 2D illustration showing how the optimal decision boundaries differ between with and without Label Smoothing. Similarly I would suggest an illustrative sketch over 3 classes (visually in the same vein as those in [1]) of how greater variance contributes to loss. I think the text could also be improved by highlighting the key theoretical takeaways in plain English (e.g. in a box/table/in bold), allowing the less mathematically inclined reader to not have to parse through the comparatively dense and notation heavy theory.
2. Missing related work: there are a few missing references to label smoothing in the context of uncertainty estimation [2,3], neural collapse [4,5] and transfer learning [6].  [4,5,6]  are especially relevant and should be discussed as they directly measure how tight features cluster together after training with label smoothing. [6] also demonstrates that tighter feature clusters lead to **worse** linear probing performance.
3. Limited insight with regards to generalisation. The paper lacks any direct insight into how label augmentation leads to better generalisation peformance, which is ultimately why these techniques are used in the first place. Although the authors do fully acknowledge this limitation, it doesn't change the fact that this limits the impact of the insights of the paper.
4. The practical takeaways of the spurious correlation experiments are somewhat unclear. It is not necessarily the case that spurious correlations tend to be lower variance. In the case that they are high variance wouldn't the inverse behaviour manifest?

[1] Muller et al. When Does Label Smoothing Help? 2019

[2] Zhu et al. Rethinking Confidence Calibration for Failure Prediction, 2022

[3] Xia et al. Towards Understanding Why Label Smoothing Degrades Selective Classification and How to Fix It, 2024

[4] Zhou et al. Are All Losses Created Equal: A Neural Collapse Perspective, 2022

[5] Guo et al. Cross Entropy versus Label Smoothing: A Neural Collapse Perspective, 2024

[6] Kornblith et al. Why Do Better Loss Functions Lead to Less Transferable Features? 2021

**Questions:**

1. For the CIFAR experiment with the spurious feature, are you training the full model or just probing a frozen ResNet with a linear logistic classifier? Additionally can you clarify the difference between line 65 where you specify "if the high variance feature satisfies a greater separability" and the experiment where the low variance spurious feature is more separable.
2. The experiments all use heavily overparameterised networks. 2048 is somewhat overkill for MNIST and ResNet-18 is not designed for CIFAR but ImageNet. ResNet-18 has over 10 million parameters whilst ResNet-56 which is designed for CIFAR has < 1 million and both achieve similar performance on CIFAR. Can you comment on how overparameterisation/overfitting affects the theory/experiments of the paper? This contrasts to the binary linear scenario where underfitting/model being insufficiently expressive may be more likely(?).

I am happy to raise my score if the stated weaknesses (other than 3 which I feel unfortunately can't be changed) and above questions are addressed.

Edit: as the authors have addressed most of my queries and concerns, and also improved the paper in response to the other reviewers, I have raised my score from 5 to 6.

---

> ### Author Response · Authors · 2024-11-18
> **Response to Reviewer Gh1c**
>
> We would like to thank Reviewer Gh1c for taking the time to review our work and for the useful feedback. We hope to address the stated concerns below.
>
> ## Weaknesses
>
> 1. We apologize for the lack of visual aids for illustrating the theoretical results, this was an oversight on our part and we completely agree that such visualizations provide a lot of practical intuition for the theory. In the revision, we now have an Appendix C.1 that visualizes decision boundaries for weight decay, label smoothing, and Mixup in a simple 2-D case. We have also restructured the main contributions section of the paper (Section 1.1) to better highlight our takeaways.
>
> 2. Thank you for pointing out these references that we missed -- we now include all of them, and have also restructured the related work section to include a discussion of the relationship to neural collapse.
>
> 3. We agree that our paper does not directly show that minimum variance feature learning leads to better generalization in standard benchmarks, as this would require clearly defining such features with respect to the benchmarks and illustrating that the high variance features are mostly noise in the training data. This type of analysis has been done to an extent in prior work [1, 2], but our goal was to take the observations of the prior work and generalize them at the cost of the setting in which we prove them. Furthermore, as we now emphasize in the main contributions section, our results do not immediately imply generalization results because sometimes the low variance features in the data can generalize poorly, as in the experiments of Section 4.2.
>
> 4. You are right, spurious correlations do not need to be low variance -- we merely wanted to illustrate that when they are, the minimum variance learning property can actually be detrimental. We hypothesize that in standard benchmarks the spurious correlations are actually high variance, and the ``good'' features are low variance.
>
> ## Questions
>
> 1. In the CIFAR experiments, there is no pretrained model being used -- we directly train a logistic regression model on the CIFAR datasets. In the setups of Section 4.2, the low variance feature (both the adversarially introduced pixels in Section 4.2.1 and the backgrounds in Section 4.2.2) have smaller magnitudes than the normalized (i.e. $z$-scored) pixels in the rest of the image (the higher variance feature(s) in this case); we intended these setups to combine both the separability and variance assumptions from our theory.
>
> 2. The overparameterization definitely affects the learning dynamics of the ResNet models, which definitely interacts with the final layer activation variances we report. However, we do not believe it affects our main takeaways (as our theory either handles linear models or is stated in a level of generality that ignores the model type), and we discuss the impact of scaling overparameterization in Appendix C.3.
>
> We hope the above changes, along with other changes pointed out in the top-level comment, serve to improve upon the weaknesses you pointed out and are happy to answer any further questions you may have.
>
> [1] Chidambaram, M., Wang, X., Wu, C., \& Ge, R. (2022). Provably Learning Diverse Features in Multi-View Data with Midpoint Mixup. International Conference on Machine Learning.
>
> [2] Zou, D., Cao, Y., Li, Y., \& Gu, Q. (2023). The Benefits of Mixup for Feature Learning. International Conference on Machine Learning.

---

> > ### Comment · Reviewer_Gh1c · 2024-11-18
> >
> > The revised submission looks promising!
> > It’s late in my time zone, but I will try to get back to the authors over the next couple of days.

---

> > > ### Comment · Reviewer_Gh1c · 2024-11-19
> > >
> > > Thanks for updating the manuscript. The improvements are promising after the authors have incorporated the various feedback from reviewers. I am mostly satisfied, however, I still have a few more questions/suggestions.
> > >
> > > 1. I think the paper would benefit from 2D visualisations on Gaussian-distributed classes, for a couple of reasons.
> > >     1. When discussing "variance" as a concept, the average reader I feel would most strongly associate it with Gaussians. Using a visual like this would quickly allow readers to intuit the first theoretical result. I would even (subjectively) suggest placing these on the first or second page for visual appeal with a general machine learning audience (although the paper is at the page limit, there is still whitespace to be worked with).
> > >     2. Relation to the true decision boundary -- when the variances are all equal the generative Gaussian classifier reduces to a logistic classifer but in general the decision boundary is quadratic. It would be interesting to see how LS, Mixup and weight decay differ not only to each other, but also to the true generative Gaussian decision boundary.
> > > 2. With regards to my question on separability. My understanding of line 203 is that you require the high variance feature to be more separable. However, it appears that the low variance (spurious) features in the experiments are actually more separable (e.g. the BG colours in MNIST are not ambiguous but there are ambiguous numbers in the dataset). I would like clarification on this apparent conflict (although this also could be my misunderstanding).
> > >
> > > > we directly train a logistic regression model on the CIFAR datasets
> > >
> > > Just to check, you train a linear model on 32x32x3 = 3072 input features?

---

> > > > ### Author Response · Authors · 2024-11-20
> > > >
> > > > Thank you for reviewing our revision and providing additional suggestions. We hope to address them below.
> > > >
> > > > 1. Regarding using Gaussian examples:
> > > >     1. We definitely agree that Gaussian distributions are a prototypical example that most readers will have familiarity with -- however, they are not the best choice in our setting due to the fact that we are working with *compactly supported* distributions in Definition 3.1 (which is an assumption modeled after image distributions). For this reason, we think the distribution that we have visualized in Appendix C.1 with variance along one dimension but no variance along the other is a better way to intuit our results. We also agree that it would be nice to have the visualizations in Appendix C.1 in the main body of the paper, but as you point out due to space constraints it is difficult to move those visualizations to the main body without significantly rearranging other aspects of the paper. While we are amenable to making this change, there are definitely tradeoffs between some audiences preferring aspects like the longer proof sketches versus having the visualizations in the main body.
> > > >
> > > >    2. Investigating the decision boundary for the augmented losses versus ERM is definitely an interesting direction for getting a better intuition for how these augmentations work, and prior works have studied this at least in the context of Mixup [1, 2]. In particular, [1] shows that Mixup and ERM have the same optimal classifier (in finite samples) with high probability on data generated from a single high-dimensional Gaussian (and labeled arbitrarily, taking advantage of the fact that $d$ is larger than $n$) and [2] shows that both ERM and Mixup align with the Bayes optimal direction in the setting of data drawn from two symmetric (i.e. means are same but different sign) Gaussians with the same covariance but that this is not true for certain Mixup variants. We are not aware of similar analysis for label smoothing, but that could be a worthwhile calculation.
> > > >
> > > > 2. Here we are thinking about separability in the sense of Assumption 3.2; even though the backgrounds in the colored MNIST example are more unambiguous than the numbers themselves, the background pixel intensity is small relative to the intensity of the numbers. Thus, intuitively one can get a larger correlation with the target at a fixed weight norm by focusing on the pixels corresponding to the numbers instead of the backgrounds, and in that sense the numbers are more separable. Similarly, in the case of the CIFAR example, the spuriously correlated pixel introduced into the training data has much lower magnitude than the rest of the image and consequently requires more weight in that dimension to achieve the same correlation that could be achieved by placing weight in other dimensions.
> > > >
> > > > And yup, for the CIFAR experiments we directly train a logistic regression model on the image data converted to flattened tensors. Our goal here was just to examine the effect of introducing the adversarial spurious correlations in the training data. We could alternatively have first run the data through a frozen model before fitting the logistic regression, but that would work out the same so long as the adversarial perturbations were applied to the features extracted from the frozen model. Basically, both label smoothing and Mixup hone in on the adversarially introduced zero class-conditional variance pixels even if we combine them with weight decay.
> > > >
> > > > [1] Chidambaram, M., Wang, X., Hu, Y., Wu, C., & Ge, R. (2021). Towards Understanding the Data Dependency of Mixup-style Training. International Conference on Learning Representations.
> > > >
> > > > [2] Oh, J., & Yun, C. (2023). Provable Benefit of Mixup for Finding Optimal Decision Boundaries. International Conference on Machine Learning.

---

> > > > > ### Comment · Reviewer_Gh1c · 2024-11-20
> > > > >
> > > > > Thanks for getting back to me so quickly. Forgive the writing as I am typing on my phone. Thank you for the clarification and the additional context from related work.
> > > > >
> > > > > 1. Thanks for clarifying, although I would like to follow up out of curiosity. Do you think the decisions boundaries for the Gaussian case will not obey the behaviour of label augmentation preferring low variance features then, since the setup in sec 3 is violated? Additionally I did struggle when revisiting sec 3 —perhaps a glossary of notation in the appendix would benefit readers?
> > > > > 2. I think I understand now what you mean by “separability”, however I would request you add a short sentence such that misunderstandings like mine are less likely to occur.
> > > > >
> > > > >
> > > > > Again, I would suggest a sentence to clarify the CIFAR experiment to avoid confusion like mine.

---

> ### Author Response · Authors · 2024-11-21
>
> Thank you for the prompt response. Here we hope to address your additional concerns.
>
> 1. The intuition we present still works, i.e. honing in on low variance dimensions/features is still important for minimizing loss since it allows the model to more closely predict the soft labels. However, when we relax the compact support constraint we can't rely on strong convexity of the augmented losses in the way that we have done to separate them from ERM. This is demonstrated by the case of Gaussians where Class -1 is $\mathcal{N}(-\mu, \Sigma)$ and Class +1 is $\mathcal{N}(\mu, \Sigma)$. Here, the Bayes decision boundary is given by $\Sigma^{-1} \mu$ (so we are weighting with respect to the inverse of the variance) and both the ERM and Mixup minimizers were shown to align with this direction by [1]. So while there isn't separation in terms of the learned direction in this case, there is still a preference for low variance effect.
>
> 2. We have modified the text so as to clear up the confusion regarding separability, and also made clear that no feature extraction is applied to the CIFAR data before we apply logistic regression.
>
> Feel free to let us know if you have any further questions or suggestions.
>
> [1] Oh, J., & Yun, C. (2023). Provable Benefit of Mixup for Finding Optimal Decision Boundaries. International Conference on Machine Learning.

---

> > ### Comment · Reviewer_Gh1c · 2024-11-21
> >
> > Thanks for clarifying the text.
> >
> > > So while there isn't separation in terms of the learned direction in this case, there is still a preference for low variance effect.
> >
> > That's interesting, so in this case there is no difference in the direction of the decision boundary regardless of whether label augmentation is used or not? Perhaps it's worth mentioning in the appendix that this scenario may be a limitation of the theory.
> >
> > I will raise my score to a 6 after this final query is resolved.

---

> > > ### Author Response · Authors · 2024-11-21
> > >
> > > No problem, and we certainly agree that emphasizing this contrast in settings is worth doing -- we have edited Appendix C.1 with a remark that now points this out. Thank you for your continued engagement, and feel free to let us know if you have other concerns.

---

> > > > ### Comment · Reviewer_Gh1c · 2024-11-22
> > > >
> > > > Thank you for adding the requested remark.
> > > >
> > > > I hope the authors feel that the quality of the paper has been improved by the review process of ICLR, I certainly think so myself.
> > > >
> > > > I have accordingly raised my score from 5 to 6.

---

### Official Review · Reviewer_2SFa · 2024-11-01

**Soundness:** 3
**Presentation:** 3
**Contribution:** 3
**Rating:** 8
**Confidence:** 5

**Summary:**

The paper presents a theoretical analysis of label smoothing and Mixup techniques. The authors investigate the differences between these label augmentation methods and the vanilla approach (i.e., empirical risk minimization with weight decay) regarding their effectiveness in learning features of varying variance. In a binary linear classification problem, they prove that, under certain assumptions, the vanilla method prefers high-variance features, while label augmentation encourages the model to learn low-variance features. This analysis is then extended to more complex settings by showing that the label augmentation loss is lower-bounded by model variance. The theoretical findings are supported by empirical results on both original and modified CIFAR-10, CIFAR-100, and colored MNIST datasets.

**Strengths:**

The theoretical results are insightful and are effectively supported by a variety of well-designed experiments.

**Weaknesses:**

- It remains unclear why learning low-variance features leads to better generalization in practical scenarios, as this approach can result in poorer generalization on synthetic datasets.
- The comparison to previous works on feature learning (e.g., Chidambaram et al., 2023; Zou et al., 2023) is insufficient. I encourage the authors to provide a more detailed comparison with these works (e.g. empirical loss vs population loss). Additionally, the notion of "features" appears to differ between this work and prior studies; clarifying what "features" specifically refers to in this context would enhance the reader’s understanding.

### Minors

- Line 795: “approach of Theorem 3.4” → "approach of Theorem 3.3" and I hope the author provides a complete proof of Corollary 3.4 for clarity, even if it largely repeats the proof of Theorem 3.3.

**Questions:**

- In the proof of Corollary 3.4, what happens if the max-margin solution does not include high-variance features? It appears that the proof technique may not apply in this scenario. Could the authors clarify whether this case is impossible or if there are additional conditions to address it?

---

> ### Author Response · Authors · 2024-11-18
> **Response to Reviewer 2SFa**
>
> We would like to thank Reviewer 2SFa for reviewing our paper, and we are grateful that they found our results to be insightful. We hope to address the pointed out weaknesses and questions below.
>
> ## Weaknesses
>
> - In our work, our goal was only to prove this minimum variance property of Label Smoothing and Mixup -- given that both methods almost always lead to performance improvements in practice, we hypothesize that learning minimum variance features is actually important for the standard datasets/benchmarks. We completely agree that this is only correlation and not causation; this is why we believe that investigating such minimum variance features in standard datasets is a promising direction for future work.
>
> - Thank you for the suggestion -- we have now added Appendix B which covers the settings of the prior works of Chidambaram et al. (2023) and Zou et al. (2023) in more detail and contrasts it to ours.
>
> ## Minor Comments and Questions
>
> - We have revised the proof of Corollary 3.4 and in the process fixed the typo, thank you for catching.
>
> - Under Assumption 3.2, the max-margin direction must have nontrivial weight in the high variance direction. This should be clearer from the added detail to Corollary 3.4.
>
> We are happy to answer any further questions you may have.

---

> ### Comment · Reviewer_2SFa · 2024-11-18
>
> Thank you for your response. I have reviewed the revised version of the paper and I am happy to see that the authors have effectively addressed my main concern. In particular, the restructured main contributions section is significantly clearer, and I believe it successfully eliminates any potential misunderstandings or confusion for readers. As a result, I am increasing my score to 8.

---

### Official Review · Reviewer_Enmj · 2024-11-02

**Soundness:** 2
**Presentation:** 3
**Contribution:** 3
**Rating:** 6
**Confidence:** 3

**Summary:**

This manuscript studies the relationship between label augmentation techniques (label smoothing and Mixup) and feature variance. The authors demonstrate that label augmentation in binary classification with a linear model encourages the model to learn low-variance features. In contrast, ERM with weight decay can lead to learning higher-variance features under certain conditions. They also show that the losses of label smoothing and Mixup for nonlinear models are lower-bounded by a function of the model output variance. Additionally, they provide empirical evidence that label augmentation may negatively impact model training.

**Strengths:**

1. This manuscript is well-written, and the theoretical parts are clearly presented.
2. This manuscript links label smoothing and Mixup to feature variance, offering new insights into the analysis of label augmentation.

**Weaknesses:**

1. The main claim that “optimizing the empirical risk with weight decay can learn higher variance features” is built on a strong assumption. As the authors note, Assumption 3.2 is indeed strong, suggesting that the $\\mathcal{H}$ dimensions are inherently better suited for classification than the $\\mathcal{L}$ dimensions. However, it’s straightforward to construct a task that relies more heavily on the $\mathcal{L}$ dimensions than on the $\mathcal{H}$ dimensions. For instance, consider a 2-D distribution where the $x$ coordinates for both classes are uniformly ranging from -1 to 1, while the $y$ coordinates are set at -0.1 or 0.1, depending on the class. In fact, by simply reversing the assumption from $y \\langle v^*, x \\rangle > y \langle u^*, x \\rangle$ to $y \\langle u^*, x \\rangle > y \langle v^*, x \rangle$, the conclusion of Theorem 3.3 changes from $\\|w^*_\\mathcal{H}\\|^2 \ge \frac{1}{2} \\|w^*\\|^2$ to $\\|w^*_\\mathcal{L}\\|^2 \ge \frac{1}{2} \\|w^*\\|^2$.
2. There is no discussion on why label smoothing or Mixup learn more on low-variance features compared to ERM or ERM with weight decay. In fact, by setting $\\alpha$ to 0, the label smoothing loss reduces to the negative log-likelihood loss, allowing one to obtain results similar to those in Theorem 3.6 or Proposition 3.10 for ERM, which also demonstrate that the ERM loss is lower-bounded by a function of the model output variance. Thus, $\\alpha$ plays a central role in learning low-variance features, but this important analysis is missing.

**Questions:**

1. Can these results extend to multi-label classification?

---

> ### Author Response · Authors · 2024-11-18
> **Response to Reviewer Enmj**
>
> We would like to thank Reviewer Enmj for taking the time to review our paper, and we appreciate that they found our results well-presented. We hope to address the raised questions below.
>
> ## Weaknesses
>
> 1. We agree and as you point out already note in the main paper that Assumption 3.2 is strong; we intended it to set the scene for showing that Mixup/Label Smoothing will still prefer the low variance feature even when it can be worse for classification than the high variance feature. We are not arguing that the high variance features are always, or even typically, the better features in practical settings -- in fact, we expect the opposite since Mixup/Label Smoothing tend to always help performance. And we also agree that, as far as Theorem 3.3 is concerned, $\mathcal{L}$ and $\mathcal{H}$ are just placeholders; we only introduce variance assumptions in Assumption 3.5. Once again, the use of $\mathcal{L}$ and $\mathcal{H}$ here even in Theorem 3.3 is to make it easier to compare the result to Theorems 3.6 and 3.7.
>
> 2. The discussion on *why* Mixup/Label Smoothing learn the low variance features but not the high variance features, whereas ERM can/will learn both is provided in detail in Sections 3 and 4, in particular by contrasting Theorem 3.2 to Theorems 3.6 and 3.7. It is **not true** that you can just take $\alpha = 0$ and prove the results we prove for Label Smoothing in these sections; that is in fact the purpose of Assumption 3.2 and Theorem 3.3 -- they are agnostic to the variance of the features. We have revised the writing in Section 3 to better emphasize this.
>
> ## Questions
>
> 1. Extending our results to the multi-label case should certainly be possible if we reduce the multi-label problem to several independent binary classification problems (determining whether a certain label should be assigned or not to an input).
>
> We are happy to address any further concerns you may have, and hope that the above clarifies our results further.

---

> > ### Comment · Reviewer_Enmj · 2024-11-24
> >
> > Thanks for the response. After re-examining the proof, I realized that the condition $\\alpha > 0$ is necessary for Theorem 3.6. I apologize for the misunderstanding in my initial review. Accordingly, I will revise my score from 5 to 6.

---

### Author Response · Authors · 2024-11-18
**Summary of Revisions**

We would like to thank all of the reviewers for their helpful feedback and suggestions. Here we summarize the main changes in the revision to the paper.

1. We have added Appendix B which provides a much more detailed comparison to the settings of the most closely prior works [1, 2].

2. We have added Appendix C.1 which provides visualizations of learned decision boundaries, to provide more intuition for our theoretical results.

3. We have added Appendix C.2 which considers experiments on synthetic data that exactly follows the assumptions in our theory, to provide a direct verification of the theory.

4. We have restructured the related work to include a discussion of related results in the neural collapse literature.

5. We have restructured the main contributions section (Section 1.1) to simplify the takeaways of our paper.

We hope to continue to engage with the reviewers throughout the discussion period.

[1] Chidambaram, M., Wang, X., Wu, C., & Ge, R. (2022). Provably Learning Diverse Features in Multi-View Data with Midpoint Mixup. International Conference on Machine Learning.

[2] Zou, D., Cao, Y., Li, Y., & Gu, Q. (2023). The Benefits of Mixup for Feature Learning. International Conference on Machine Learning.

---

### Meta-Review · Area_Chair_ruv4 · 2024-12-22

**Metareview:**

The paper studies the effect of label manipulation in classification tasks. It achieves formal results on the effect of two common manipulations, label smoothing and mixup, on the variance of features that are learnt for linear binary classifiers. Then, it studies non-linear models with the same perspective and derives bounds for the model’s output in presence of label smoothing and mixup. A remark of their formal study is that while such label manipulations can be effective for generalization it can potentially make them more prone to learning and predicting based on unwanted correlations. It empirically shows that this is indeed the case.

The reviewers acknowledged the insights the paper offers and the clarity of the writing.

On the other hand, the reviewers criticized several aspects of the formal results including the assumptions, their relevance including a discussion of causality, the notations, and the connections between the different formal statements. They also raised concerns regarding the coverage of prior works particularly on the topics of feature learning and label smoothing.

The authors provided a rebuttal to those points mostly including clarifications regarding their formal results where they also updated the paper accordingly. They further included the missing references in the revised paper.

The reviewers, after discussion with the authors, generally lean towards acceptance.

The AC does not find solid reason to go against the reviewer’s high majority rating and recommends acceptance. But as discussed with 2SFa and others it is important for the paper to repeatedly emphasize that (1) the relation between feature variance and generalization is not studied in this work and a causality is not to be assumed, and (2) low variance features is not shown to be particularly worse for spurious correlations compared to high-variance features.

**Additional Comments On Reviewer Discussion:**

Five reviewers evaluated the paper with relevant expertise such as label smoothing, mixup, and learning dynamics. Reviewers are eventually mostly at the borderline with the exception of one reviewer that suggests clear acceptance. Three out of four borderline reviewers lean towards acceptance after discussion with the authors.

---

### Decision · Program_Chairs · 2025-01-22

Accept (Poster)